# INDUCTION HEAD IMPLEMENTATION ACROSS DIVERSE TRANSFORMER WEIGHT CONSTRUCTIONS

## ABSTRACT

Induction heads are a class of self-attention mechanisms empirically crucial for in-context learning in transformer models. Although previous research has suggested possible forms of induction heads, it remains unclear how they interact with other network modules and operate on their outputs. In this work, we address this question by showing that a two-layer induction head allows flexibility in the construction of its first layer. This flexibility enables the induction head to operate alongside other modules within the network. Additionally, in the multi-layer networks where it's difficult for the induction heads to retrieve the original input, we propose a new mechanism akin to induction heads (in the sense of using the information of inter-token identity) that still functions in deep networks. We also performed proof-of-concept experiments showing that induction heads are trainable using only a subset of the model's layers. Our key insight is that information about which tokens are identical are possible to be extracted from the outputs of many transformer networks, which is essential for applying the induction head mechanism. Our work presents the diversity in the realization of induction heads, which serves as an explanation for why induction heads consistently appear across models.

## 1 INTRODUCTION

In modern transformer-based language models, induction heads (Olsson et al., 2022) have emerged as a fundamental mechanism underlying in-context learning: they enable a model to detect repeated token patterns in its input and use them to predict subsequent tokens, effectively "completing" sequences of the form "$\ldots A\,B \ldots A \rightarrow B$", where $A$ and $B$ denote arbitrary token substrings. This match-and-copy behavior is implemented by a specialized pair of attention heads across layers, which first identify prior occurrences of $A$ and then attend to the corresponding $B$ tokens at prediction time.

Previous theoretical works on induction heads includes discussions on the possible formation of such heads and the training dynamics to attain them (Chen et al., 2024; Wang et al., 2024; Sanford et al., 2024). In which, Chen et al. (2024) provide the first provable characterization of how full transformer architectures (including attention, positional embeddings, feed-forward layers, and normalization) jointly train to implement the induction head mechanism for in-context learning on synthetic $n$-gram data. Wang et al. (2024) follow the route of Chen et al. (2024) and gives more efficient constructions as well as tracking training trajectories to reveal the phase transition in trainings on $n$-gram data. Sanford et al. (2024) on the other hand, demonstrate a single-layer of transformer is not able to approximate an induction head efficiently.

Comparing to prior research, our work establishes the theory of transformer-approximated induction heads both using a different method and delivering a different perspective. Our work not only proves transformer has the capability of approximating induction heads, but also shows that this job can be accomplished by a wide range of transformers. Most importantly, our work shows that

> Induction heads are not rigidly defined modules with a fixed design, but adaptable components that may merge into networks of different purposes.

Our work also technically stand out by removing the reliance of relative positional embedding (RPE) that's crucial to previous work (Chen et al., 2024; Wang et al., 2024). Instead, our result applies to all full-rank positional encodings matrices. Furthermore, we develop a theoretical framework explaining

how induction heads may be realized in multi-layer (more than 2) networks. We demonstrate that information about token identity — specifically, which tokens match and which do not — remains robust under the transformer's successive transformations and provides feasibility for multi-layer networks to implement induction heads.

**Contributions.** Our contributions are summarized as follows.

- We adopt a new theoretical approach to show that transformers has the capability of approximating induction heads, allowing us to remove the reliance on relative positional embedding (RPE) in previous research.
- We demonstrate that applying the induction does not require a specific construction of transformers. In fact, We show that a substantial portion of an induction head's weights can be arbitrary. This finding provides a possible explanation for why induction heads emerge across different models. We further validate this by experiments demonstrating that induction heads can still be trained even when a significant part of the network is randomly fixed.
- We propose a theoretical framework for the implementation of induction heads on multi-layer networks.

Our findings establish induction heads not as a function of specific models circuits, but as a general feature across diverse transformer architectures. This perspective both consolidates prior empirical findings and provide technical insight for future theoretical analyses.

RELATED WORKS

Induction heads are specialized attention heads that implement a copy-paste, pattern-matching algorithm (e.g., matching "[A][B] ... [A]" to predict "[B]") (Elhage et al., 2021). They have been identified as a key mechanism for in-context learning (ICL) in transformers. Mechanistic analyses first showed that these heads emerge abruptly during training, coinciding with sudden jumps in ICL performance, suggesting a phase transition in training dynamics (Olsson et al., 2022; Chen et al., 2024). Olsson et al. (2022) provide preliminary empirical evidence linking induction-head formation to sharp decreases in loss and improved few-shot generalization, proposing induction circuits as foundational for ICL.

Subsequent works formalized this phenomenon: for example, Chen et al. (2024) prove that training a two-layer transformer on synthetic $n$-gram data converges under gradient flow to a model implementing the induction-head mechanism. Other theoretical analyses of simplified transformers similarly found that induction-like attention patterns naturally arise in shallow models trained on Markovian or algorithmic tasks (Nichani et al., 2024).

Further empirical studies reinforced the centrality of induction heads. Ablating these heads in small- to medium-scale language models significantly degrades ICL performance. For instance, removing a few induction heads in 3B–20B parameter LMs decreases few-shot accuracy by approximately 30%, reducing performance to near chance levels (Crosbie & Shutova, 2024). However, recent interpretability results indicate that induction heads dominate ICL primarily in smaller and earlier-stage models. Larger models shift toward latent representation-based mechanisms, particularly function-vector heads, which encode task-specific vectors instead of copying tokens (Yin & Steinhardt, 2025). Many such latent heads initially behave like induction heads before transitioning to function-vector strategies, suggesting induction heads bootstrap more complex in-context representations.

Notably, previous studies of induction head have often focused on a specific construction of circuit and the training dynamic to obtain it. Indeed, the prevalence of induction heads varies across architectures, raising questions about the universality of these mechanisms. In contrast, our work demonstrates the remarkable universality of induction heads, showing their emergence across diverse transformer architectures and providing analytic characterizations extending beyond the canonical decoder-only setting.

## 2    PRELIMINARIES AND PROBLEM SETUP

**Notations.**    For a vector $v$, we denote its $i$-th entry by $v_i$ and its subvector from the $i_1$-th to the $i_2$-th entry (inclusive) by $v_{i_1:i_2}$ with $i_1 < i_2$. For a matrix $M$, we use $M_{i,j}$ for the entry in the $i$-th row and $j$-th column, $M_{i,:}$ for the $i$-th row, and $M_{:,j}$ for the $j$-th column. We also use $M_j$ to denote the $j$-th

column of the matrix $M$ when it does not cause confusion. The submatrix spanning rows $i_1$ through $i_2$ and columns $j_1$ through $j_2$ is denoted by $M_{i_1:i_2,,j_1:j_2}$ with $i_1 < i_2,, j_1 < j_2$. We define $c_{a \times b}$ as an $a \times b$ matrix with constant entries $c$, and abbreviate $c_{a \times 1}$ as $c_a$. For norms, we define $\| \cdot \|_\infty$ as the maximum absolute element in a vector or matrix. The $p$-norms are given by $\|v\|_p = (\sum_i |v_i|^p)^{1/p}$ for a vector $v$ and $\|M\|_p = (\sum_{i,j} |M_{i,j}|^p)^{1/p}$ for a matrix $M$. We define the operator norm of matrix as

$$M_{\alpha,\beta} = \sup_{\|x\|_\alpha \leq 1} (\|Ax\|_\beta).$$

For function norms, we define the $L_\infty$ norm as $\|f\|_{L_\infty} := \sup_{x \in X_f} \|f(x)\|_\infty$, where $X_f$ is the input domain of $f$. For functions, when a function $f : \mathbb{R} \to \mathbb{R}$ is applied on a vector or a matrix, it means to apply $f$ on every entry of the vector/matrix (i.e., $\exp([a_1, a_2]) := [\exp(a_1), \exp(a_2)]$).

**Indicator of Identity.**    We define the following indicator of identity that is key to the implementation of induction head.

**Definition 2.1** (Indicator of Identity). For an input sequence

$$X = \begin{bmatrix} x_1 & x_2 & \cdots & x_n. \end{bmatrix}$$

Define $\mathrm{ID}(X) : \mathbb{R}^{d \times n} \to \mathbb{R}^n$ as

$$\mathrm{ID}(X) := \begin{bmatrix} 0 & \mathbb{1}_{x_1 = x_n} & \mathbb{1}_{x_2 = x_n} \cdots & \mathbb{1}_{x_{n-1} = x_n} \end{bmatrix}^\top,$$

where $\mathbb{1}_{x_i = x_n}$ is

$$\mathbb{1}_{x_i = x_n} = \begin{cases} 1, & x_i = x_n, \\ 0, & x_i \neq x_n. \end{cases}$$

**Dictionary.**    We then define the dictionary from which the tokens are sampled.

**Definition 2.2** (Dictionary). We define dictionary $\mathcal{D}$ of token dimension $d$ as the set of all tokens. All tokens in the input sequence come from the dictionary.

**Self-Attention Mechanism.**    For an input sequence in $\mathbb{R}^{d_i}$, where $d_i$ is the input dimension, we use SA to denote a self attention block defined as

$$\mathrm{SA}(Z) := W_V Z \, \mathrm{Softmax}((W_K Z)^\top W_Q Z),$$

where $W_V \in \mathbb{R}^{d_i \times d_i}, W_K, W_Q \in \mathbb{R}^{d_i \times d_h}$ are its parameters. Here $d_h$ denotes the hidden dimension of the attention block.

For an $H$-head multi-head attentions (denoted as MA), we adopt the following definition similar to that in (Chen et al., 2024),

$$\mathrm{MA}(Z) := \mathrm{Concat}(\mathrm{SA}_1(Z), \mathrm{SA}_2(Z), \cdots, \mathrm{SA}_H(Z)) \in \mathbb{R}^{H d_i \times n},$$

where $\mathrm{SA}_i$ denotes the $i$-th head of MA, and Concat means concatenating the output of $\mathrm{SA}_i$ in the column direction (expanding the token-length).

In addition, we define transformer $T$ as an attention followed by a feed-forward network

$$T := \mathrm{SA} \circ \mathrm{FFN}.$$

**Induction Head.**    We define the induction head as follows.

**Definition 2.3** (Induction Head). Let $X$ denote the input sequence and $x_i$, $i \in [n]$ denote its tokens, written as

$$X := \begin{bmatrix} x_1 & x_2 & \cdots & x_n \end{bmatrix}.$$

Let $D_n$ denote all labels of the tokens in $X$ that are identical to $x_n$, written as

$$D_n := \{i \mid x_i = x_n, i \neq n\}.$$

When there's only one token identical to $x_n$ in the whole sequence (except $x_n$), we define $\mathrm{IND}(X)$ as the token next to that token. Thus, we omit this situation.

When there are multiple tokens identical to $x_n$, we define the induction head $\mathrm{IND}(X)$ as the average of the tokens next to the ones in $D_n$, which is

$$\mathrm{IND}(X) := \frac{1}{|D_n|} \sum_{i \in D_n} x_{i+1},$$

where $|D_n|$ is the cardinality of $D_n$.

**Remark 2.1** (Situation of $x_i \neq x_n$ for all $i < n$)**.** When there's no token identical to $x_n$, $\mathrm{IND}(X)$ does not function and it becomes meaningless to discuss approximating it with a transformer. In this sense, we **only consider approximating an induction head on the input domains where it functions** (i.e., inputs with tokens identical to the last one).

## 3 METHOD

In this section we demonstrate our main results that a wide range of transformers are capable of implementing induction heads. Additionally, we analyses the possible form of induction head in multi-layer networks in Section 3.4 at the end of this section.

### 3.1 TRANSFORMER RECOGNIZE IDENTITY

The following assumption of dictionary is applied to all our results, which states the dictionary where tokens are sampled from must be finite and must exclude the all-zero token.

**Assumption 3.1** (Assumption of Dictionary)**.** For dictionary $\mathcal{D}$, assume

$$|\mathcal{D}| < \infty \quad \text{and} \quad 0_d \notin \mathcal{D}.$$

We then prove a single layer of transformer is able to implement the indicator in Definition A.1. Furthermore, we observe that the construction of this transformer is very flexible. We state present result as Corollary A.1.1.

**Theorem 3.1** (Transformer Recognize Identity)**.** For a Dictionary $\mathcal{D}$ satisfying Assumption 3.1, let

$$X := [x_1 \quad x_2 \quad \cdots \quad x_n] \quad \text{with} \quad x_i \in \mathcal{D}, \quad \text{for} \quad i \in [n],$$

denote the input sequence. And let $X_p$ denote $X$ with any full-rank positional embedding $R \in \mathbb{R}^{n \times n}$

$$X_p := \begin{bmatrix} X \\ R \end{bmatrix} \quad \text{with every column denoted as} \quad [x_1^p \quad x_2^p \quad \cdots \quad x_n^p].$$

Let $W_V \in \mathbb{R}^{(d+n) \times (d+n)}, W_K, W_Q \in \mathbb{R}^{d_h \times (d+n)}$ denote the parameters of a single-head attention SA, where $d_h \in \mathbb{N}_+$ is the number of the hidden dimension. We assume $d_h \geq d$ (hidden dimension is no less than the input dimension). Then there exists a one-layer feed-forward network FFN and a self-attention SA such that the output satisfies

$$\mathrm{FFN} \circ \mathrm{SA}(X_p) = [\mathrm{ID}(X) \quad \mathrm{ID}(X) \quad \cdots \quad \mathrm{ID}(X)].$$

*Proof Sketch.* The proof of the above theorem consists of the following steps.

- **Step 1.** Use SA to calculate the attention score matrix (i.e., $\mathrm{Softmax}((W_K X_p)^\top (W_Q X)))$. With few conditions on SA's parameter, every column $c$ of this score matrix suffices that

$$c_i = c_j \iff x_i = x_j, \quad i,j \in [n].$$

- **Step 2.** Use linear transformation to retrieve the attention score matrix from the output of SA.

- **Step 3.** Use FFN to extract the entries in the columns of the attention score matrix that is identical to the $n$-th entry. Then, map the extracted entries to $1$ and others to $0$.

- **Step 4.** Use FFN to shift the output in **Step 3** down by $1$ row and fill the first row with all $0$.

□

See detailed proof in Theorem A.1.

**Corollary 3.1.1** (Choice of Attention). Define

$$W_K := [W_K^* \quad 0_{d_h \times n}],$$

where $W_K^*$ is any full-rank matrix (in the sense of row rank (if $d_h \leq d$) or column rank (if $d_h > d$)). Let $W_Q$ be almost any matrix in $\mathbb{R}^{d_h \times (d+n)}$ (excluding a zero measure set). Let $W_V$ be any full rank matrix in $\mathbb{R}^{(d+n) \times (d+n)}$. Then for an attention SA parameterized by $W_K, W_Q, W_V$, there exists an FFN such that

$$\text{FFN} \circ \text{SA}(X_p) = [\text{ID}(X) \quad \text{ID}(X) \quad \cdots \quad \text{ID}(X)]$$

## 3.2 IMPLEMENTATION OF INDUCTION HEAD BY A WIDE RANGE OF SINGLE-HEAD TRANSFORMER

We show in the following theorem that a wide range of single-head transformers are capable of applying the induction head.

**Theorem 3.2** (Two-layer, Single-Head Attention Realizes the Induction Head). Let $\mathcal{D}$ satisfy Assumption 3.1. Let $X = [x_1, \ldots, x_n] \in \mathbb{R}^{d \times n}$ and let $R \in \mathbb{R}^{n \times n}$ be any full-rank positional encoding. Define the augmented input

$$X_p := \begin{bmatrix} X \\ R \end{bmatrix} \in \mathbb{R}^{(d+n) \times n}.$$

Fix a hidden dimension $d_h \geq d$ and choose

$$W_K' := [W_K^*, 0_{d_h \times n}], \quad W_K^* \in \mathbb{R}^{d_h \times d} \text{ full column rank}, \quad \text{with} \quad d_h \geq d.$$

Let $W_Q' \in \mathbb{R}^{d_h \times (d+n)}$ be generic (outside a measure-zero set) and let $W_V' \in \mathbb{R}^{(d+n) \times (d+n)}$ be full rank. Let $\text{SA}_1$ denote single-head self-attention with parameters $(W_Q', W_K', W_V')$ acting on column sequences. Let $A_c \in \mathbb{R}^n$ be a column-wise linear readout satisfying

$$1_{1 \times n} \cdot A_c = 1.$$

Then there exist a feed-forward map $\text{FFN} : \mathbb{R}^{(d+n) \times n} \to \mathbb{R}^{(d+n) \times n}$ and a single-head self-attention $\text{SA}_2$ such that, for any $\epsilon > 0$,

$$\|\text{SA}_2(\text{FFN} \circ \text{SA}_1(X_p) + X_p) \cdot A_c - \text{IND}(X)\|_\infty \leq \epsilon,$$

whenever at least one token $x_i$ with $i \neq n$ equals $x_n$.

**Remark 3.1** (A Key Difference with Previous Work). The construction of the induction head in previous works have relied on relative positional embedding (Chen et al., 2024; Wang et al., 2024). Specifically, the attention score matrix in their first layer is **only dependent** on the relative positional embedding. While in our work, the attention score matrix in their first layer **partially excludes** the positional embedding as part of $W_K$ multiplied on the positional encoding is set to be all 0. We'd like to denote that this is a key difference between our methodology.

The detailed proof for this theorem is in Theorem A.2.

**Proof Sketch.** Here we provide proof sketch to highlight our techniques and intuition.

We construct two single–head layers.

- **Layer 1: Indicator Extractor.** This single-head attention layer is constructed as in Theorem 3.1. Consequently, it outputs the same $\text{ID}(X)$ function for every column, which indicates the tokens that are identical to the last one (by its non-zero entries).
- **Layer 2: Token Duplication.** This single-head attention layer is constructed to duplicate the tokens in the original input sequence based on $\text{ID}(X)$ from the output of Layer 1. Specifically,

Layer 2 duplicates the token next to the token indicated by $\mathrm{ID}(X)$ (if multiple identity exists, take the average or take the last one). Since $\mathrm{ID}(X)$ indicates the tokens that are identical to the last one, Layer 2 outputs the tokens next to the one identical to the last one, which is the output of an induction head.

Thus two layers suffice to approximate the induction head arbitrarily well.

### 3.3 IMPLEMENTATION OF INDUCTION HEAD BY A WIDE RANGE OF MULTI-HEAD TRANSFORMERS

We also demonstrate a wide range of multi-head transformer is capable of implementing induction head in the following theorem. The proof technique for this theorem is analogous to that used in Theorem 3.2.

**Theorem 3.3** (Multi-Head Attention Followed by Single-Head Attention Applies Induction Head).
Let $X \in \mathbb{R}^{d \times n}$ denote the input sequence whose tokens are sampled from dictionary $\mathcal{D}$ satisfying Assumption 3.1. Let $X_p \in \mathbb{R}^{(d+n) \times n}$ denote the positional encoded version of $X$ and let $R \in \mathbb{R}^{n \times n}$ denote this positional encoding. Additionally, we assume $R$ to be a full-rank matrix.
Let MA be a multi-head attention and let $\mathrm{SA}_i, i \in [H]$ denote its $H$ heads. Let $W_K^i, W_Q^i \in \mathbb{R}^{(d+n)}$, $W_V^i$ denote the parameters of $\mathrm{SA}_i$.
If there exists an $i_0 \in [H]$, such that $W_K^{i_0} = [W_K^* \ 0_{d_h \times n}]$, $W_K^*$ is full-rank (in the sense of column rank), $W_Q^{i_0} \in \mathbb{R}^{d_h \times (d+n)}$ isn't in a specific set of zero measure, and $W_V \in \mathbb{R}^{(d+n) \times (d+n)}$ is full-rank, then there exists a feed-forward network FFN and a self-attention SA such that

$$\|\mathrm{SA} \circ \mathrm{FFN} \circ \mathrm{MA}(X_p)A_c - \mathrm{IND}(X)\|_\infty \le \epsilon,$$

for any $\epsilon > 0$. Here $A_c \in \mathbb{R}^n$ is any vector whose entries are positive and sum up to be one.

See Theorem A.3 for a detailed proof.

### 3.4 INDUCTION HEAD IN THE LATENT SPACE

We now discuss the possible form of induction heads in multi-layer attention networks.

In multi-layer attention networks, because the input has gone through multiple transformations, it becomes difficult to retrieve the original input. However, we observe that the identity between input tokens is still possible to be recognized from the output of the multi-layer transformation. Therefore, the information of token identity is able to influence the result of these multi-layer networks by another mechanism similar to induction heads. We call this mechanism the latent space induction head and define it as follows.

**Definition 3.1** (Latent Space Induction Head). Let $X := [x_1, x_2, \cdots, x_n]$ denote the input sequence and let $X_p$ denote the positional encoded version of $X$. Same as Definition A.3, let $D_n$ denote the label (subscript) of all tokens identical to $x_n$. Let $f$ be a model that takes $X$ as the input.
We then define the latent induction head (denoted by LaIND) as

$$\mathrm{LaIND}_f(X_p) := \frac{1}{|D_n|} \sum_{i \in D_n} f(X_p)_{i+1}.$$

When there is only one $x_i$, $i \le n - 1$ identical to $x_n$, latent space induction head is a mechanism that outputs the $i + 1$-th token of $f(X)$.

**Remark 3.2** (Intuition of Latent Space Induction Head). Compared with induction heads, which output the token that is next to the token identical to $x_n$, latent space induction heads output the processed version of the token next to that token identical to $x_n$.

**Remark 3.3** (Remark on $D_n$). Because when positionally encoded, every token is not identical to any other token. Therefore $D_n$ is not based on $X_p$ but $X$, because basing on the former option makes its definition meaningless.

We apply below assumption in the following discussion.

**Assumption 3.2** (Equal Norm of Input Tokens). We assume the input tokens ($x_i$ in $X$) to have the same 2-norm (denoted as $r_x$). This writes out as

$$\|x_i\|_2 = r_x, \quad i \in [n].$$

**Remark 3.4** (Practical Perspective). Assumption 3.2 is often able to be acquired by applying layernorm to the input sequence. This practice is also adopted in previous work (Chen et al., 2024).

We also assume the $W_V$ to satisfy the following assumption.

**Assumption 3.3** (Assumption of $W_V$). We assume $W_V \in \mathbb{R}^{(d+n) \times (d+n)}$ to have the form of

$$\begin{bmatrix} W_1 & 0_{d \times n} \\ 0_{n \times d} & W_2 \end{bmatrix},$$

where $W_2 \in \mathbb{R}^{n \times n}$ suffices $\|W_2\|_{1,1} \leq 1$. Additionally, $W_1 \in \mathbb{R}^{d \times d}$ and $W_2$ are full rank matrices.

This assumption implies that $W_V$ does not mix the first $d$ rows in the input with the last $n$ rows.

Under the above assumption, we now give our result on the induction head in multi-layer networks.

**Theorem 3.4** (Latent Space Induction Head of Multi-Layer Single-Head Attention Network with Skip-Connection). Let $X$ denote the input sequence satisfying Assumption 3.2 and let $X_p$ be the positional encoded version of it of positional encoding $I_n$. Let $F$ be any $N$-layer network of single-head attention whose $W_V$ matrices satisfy Assumption 3.3. Let $\theta_f$ denote the parameter of all $W_K$ and $W_Q$ matrices in $F$ and let $d_F$ denote its dimension.
For any $\delta, \epsilon > 0$, $N \in \mathbb{N}^+$ and any $\theta_F \in \mathbb{R}^{d_F}$ except for a subset in $\mathbb{R}^{d_f}$ of arbitrarily small measure, there exists a single-head attentions $\mathrm{SA}_1$ and two single-head transformers $T_1, T_2$ such that

$$\|T_2 \circ T_1(F \circ \mathrm{SA}_1(X_p) + X_p)A_c - \mathrm{LaIND}_{F \circ \mathrm{SA}_1 + I}(X_p)\|_\infty \leq \epsilon,$$
$$\|\mathrm{SA}_1(X_p)_{1:d,:} - X\|_\infty \leq \delta,$$

where $A_c \in \mathbb{R}^n$ is any vector whose entries are positive and sum up to one. The second condition on $\mathrm{SA}_1$ means $\mathrm{SA}_1$ almost only changes the positional encoding.

**Remark 3.5** (The Role of $\mathrm{SA}_1$). As mentioned in Theorem 3.4, $\mathrm{SA}_1$ almost doesn't alter the input tokens. In fact, it only affects the positional encodings by taking an average of the positional encodings under identical tokens, and therefore doesn't make $F$ unable to attend to the input tokens. This also means if $X$ does not contain identical token pairs, $\mathrm{SA}_1$ almost has no effect on $X_p$.

The detailed proof is in Theorem A.4.

## 4 EXPERIMENTAL STUDIES

In our main result Theorem 3.2, we prove that induction head can be implemented by a wide range of transformer networks even when part of its weights are randomly fixed.

In this section, we provide a proof-of-concept experiment for this result. We examine whether induction head can be trained when part of a network is randomly fixed. Specifically, we conduct the following experiment.

We randomly initialize a transformer network and **only train the first and fourth transformer layer** (along with the one-layer input embedding and output FFN) in the network on a synthesized dataset for training induction head.

**Objective.** Empirically validate the claim in Theorem 3.2 and observe if induction head can rise from models whose part of the weights are fixed.

**Data.** We randomly synthesized a dataset of 1500000 sequence of length 20 and vocab size 10 with repeated pattern of length 4. The final 3 tokens of the sequence will be the first 3 tokens in the pattern and the target is the last token in the pattern. 80% of the dataset is used as train set and 20% as evaluation set. The dataset is re-synthesized before each round of the experiment.

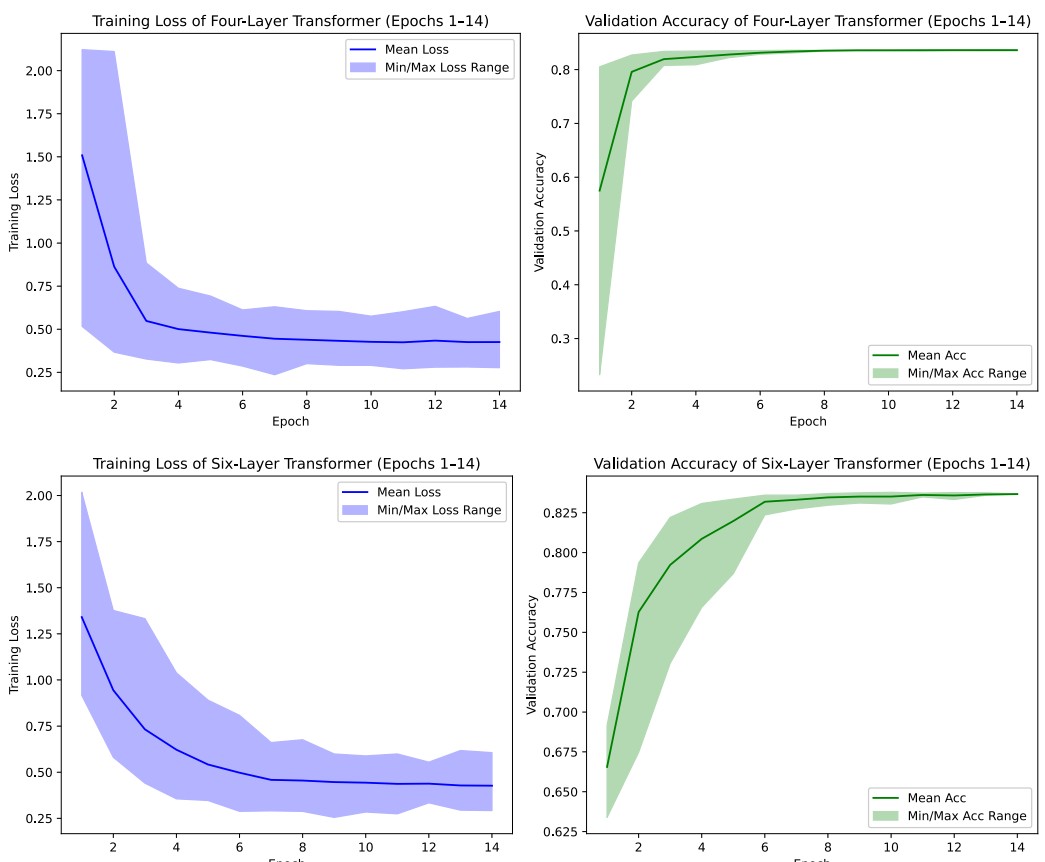

Figure 1: **The Training Losses and Validation Accuracies of Experiments.** The maximum, minimum and mean of the training loss and evaluation accuracy in 4 rounds of training on a four-layer transformer and a six-layer transformer.

**Model.** We use a four-layer and a six-layer transformer with an embedding layer (linear transformation) and a one-layer output FFN. The hidden-dimension is 400 and the number of heads is 4.

**Results.** We have run 4 rounds of experiment on both transformer network with 15 epochs of training in each round. In every round, the model reached a performance of more than 83% of accuracy on the evaluation dataset, which is around 20% when the training started. The detailed result is presented in Figure 1. The first epoch of each experiment is not presented because the loss is overly large and make results in other epochs difficult to examine.

For more experimental studies, please see Appendix C.

## 5 CONCLUSION

We establish a new theoretical understanding of induction heads in transformer architectures. Specifically, we show that they are not monolithic, rigidly designed modules but rather flexible mechanisms that can be approximated across a wide spectrum of model configurations. By removing the reliance on specific constructions, we show that induction heads emerge naturally in a wide range of transformers whose components satisfy broad conditions. Furthermore, we propose the concept of the latent space induction head to analyze how multi-layer networks may retain and propagate the information necessary for match-and-copy. This offers a possible explanation for the induction heads implemented by multi-layer networks. We conduct experiments to back up our theory in the supplement material.

**Key Insights.**    We highlight three key insights from our theoretical results:

- **Generic Construction.** We introduce a novel proof technique that bypasses the need for specific positional encodings and proves that *any* full-rank positional encoding suffices for inducing the match-and-copy behavior characteristic of induction heads.
- **Model-Agnostic Realization.** We demonstrate that induction heads do not require a narrowly tailored circuit. Instead, a broad class of transformer weight configurations—subject only to mild form and rank conditions—may give rise to the same in-context learning mechanism.
- **Multi-Layer Framework.** We develop and analyze a multi-layer theoretical framework showing that the identity-matching signals may survive successive attention transformations, thereby enabling induction heads to function in multi-layer transformer models.

These results shift the perspective on induction heads from isolated, handcrafted modules to emergent phenomena intrinsic to transformer architectures under very general conditions. By highlighting the adaptability of induction heads, our theory not only consolidates previous empirical findings but also provides a foundation for exploring how other in-context learning behaviors might similarly arise.

**Limitations.**    Our analysis rests on several idealized assumptions: a finite, non-zero dictionary of tokens, full-rank positional encodings, and fixed-norm inputs with block-diagonal $W_V$ matrices (Assumptions 3.1 to 3.3). These conditions exclude common sinusoidal embeddings, open-vocabulary settings, and networks that mix content and position more freely. The proofs address single-head layers exactly and multi-head stacks only when one head satisfies the same structural constraints. They do not cover attention with learned query–key mixing or low-rank $W_V$. All results are asymptotic and noise-free, so optimization dynamics, finite-precision effects, and pretraining data biases remain untested. Finally, the work is purely theoretical; confirming whether large language models actually realize the proposed constructions is left to future empirical study.

**Future Directions.**    Here we sketch three future directions:

- **Empirical Validation in Large-Scale Models.** While our theory applies to idealized settings, it will be valuable to empirically probe how closely practical transformer weights trained on real data align with our theoretical constructions.
- **Extensions to Other In-Context Mechanisms.** Building on our framework, one can investigate whether additional in-context capabilities—such as arithmetic reasoning or pattern extrapolation—also emerge under similar general conditions.
- **Robustness and Stability Analysis.** Understanding how noise, optimization dynamics, and architecture variations affect the formation and reliability of induction heads could yield insights for more stable and interpretable transformer designs.

By demonstrating that induction heads are broadly realizable and resilient to architectural changes, our research suggests the adoptability of induction heads to networks of diverse purpose and structure.

The generic construction of our result also makes it possible to be incorporated into other constructions of task-specific attentions in future studies. The joint function of induction head and other task-specific modules may also breed novel constructions of practical usage.

## REFERENCES

Siyu Chen, Heejune Sheen, Tianhao Wang, and Zhuoran Yang. Unveiling induction heads: Provable training dynamics and feature learning in transformers. *arXiv preprint arXiv:2409.10559*, 2024.

Joy Crosbie and Ekaterina Shutova. Induction heads as an essential mechanism for pattern matching in in-context learning. *arXiv preprint arXiv:2407.07011*, 2024.

Nelson Elhage, Neel Nanda, Catherine Olsson, Tom Henighan, Nicholas Joseph, Ben Mann, Amanda Askell, Yuntao Bai, Anna Chen, Tom Conerly, et al. A mathematical framework for transformer circuits. *Transformer Circuits Thread*, 1(1):12, 2021.

Eshaan Nichani, Alex Damian, and Jason D Lee. How transformers learn causal structure with gradient descent. *arXiv preprint arXiv:2402.14735*, 2024.

Catherine Olsson, Nelson Elhage, Neel Nanda, Nicholas Joseph, Nova DasSarma, Tom Henighan, Ben Mann, Amanda Askell, Yuntao Bai, Anna Chen, et al. In-context learning and induction heads. *arXiv preprint arXiv:2209.11895*, 2022.

Clayton Sanford, Daniel Hsu, and Matus Telgarsky. One-layer transformers fail to solve the induction heads task. *arXiv preprint arXiv:2408.14332*, 2024.

Mingze Wang, Ruoxi Yu, Lei Wu, et al. How transformers implement induction heads: Approximation and optimization analysis. *arXiv preprint arXiv:2410.11474*, 2024.

Kayo Yin and Jacob Steinhardt. Which attention heads matter for in-context learning? *arXiv preprint arXiv:2502.14010*, 2025.

# Appendix

## ETHIC STATEMENT

This paper does not involve human subjects, personally identifiable data, or sensitive applications. We do not foresee direct ethical risks. We follow the ICLR Code of Ethics and affirm that all aspects of this research comply with the principles of fairness, transparency, and integrity.

## REPRODUCIBILITY STATEMENT

We ensure reproducibility of our theoretical results by including all formal assumptions, definitions, and complete proofs in the appendix. The main text states each theorem clearly and refers to the detailed proofs. For experiments, we describe model architectures, datasets, preprocessing steps, hyperparameters, and training details in the main text. Code and scripts are provided in the supplementary materials to replicate the empirical results.

## IMPACT STATEMENT

By showing that induction-style copy mechanisms emerge under broad architectural conditions, our theory clarifies why in-context learning is ubiquitous and suggests new routes for *interpretability* and *pruning*. The results may guide safer model design: engineers can enforce or disable match-and-copy behavior by adjusting positional encodings and value pathways. Because we introduce no new data, training recipe, or deployable model, immediate misuse risk is low; the main societal effect is a deeper understanding of transformer internals that can inform both capability research and alignment efforts.

## A  PROOFS OF RESULTS IN SECTION 3

### A.1  PROOFS OF RESULT IN SECTION 3.1

In this paper, the identity between the previous token and the last token is presented in the following way.

**Definition A.1** (Indicator of Identity). For an input sequence

$$X = \begin{bmatrix} x_1 & x_2 & \cdots & x_n \end{bmatrix}$$

Define $\mathrm{ID}(X) : \mathbb{R}^{d \times n} \to \mathbb{R}^n$ as

$$\mathrm{ID}(X) := \begin{bmatrix} 0 \\ \mathbb{1}_{x_1 = x_n} \\ \mathbb{1}_{x_2 = x_n} \\ \vdots \\ \mathbb{1}_{x_{n-1} = x_n} \end{bmatrix},$$

where $\mathbb{1}_{x_i = x_n}$ is

$$\mathbb{1}_{x_i = x_n} = \begin{cases} 1, & x_i = x_n, \\ 0, & x_i \neq x_n. \end{cases}$$

We assume the tokens in the sequence are sampled from a dictionary with finite words and non all-zero token.

**Definition A.2** (Dictionary). We define dictionary $\mathcal{D}$ of token dimension $d$ as the set of all tokens. All tokens in the input sequence come from the dictionary.

**Assumption A.1** (Assumption of Dictionary). For dictionary $\mathcal{D}$, assume

$$\mathrm{card}(\mathcal{D}) < \infty$$
$$0_d \notin \mathcal{D}.$$

We give a tensor trick used in later proofs.

**Lemma A.1** (Tensor Trick). For any vector $x \in \mathbb{R}^{d_1}$, $y \in \mathbb{R}^{d_2}$ and $M \in \mathbb{R}^{d_1 \times d_2}$,

$$x^\top M y = \mathrm{Sum}(M \odot xy^\top).$$

Here $\mathrm{Sum}$ means summing all the entries in the given matrix.

*Proof.* From

$$(x^\top M y)_{i,j} = x_i M_{i,j} y_j = M_{i,j} x_i y_j,$$

and

$$(xy^\top)_{i,j} = x_i y_j,$$

we have

$$x^\top M y = M \odot xy^\top.$$

This completes the proof. $\qquad\square$

**Lemma A.2** (Almost All Attention Preserves Identity). Let $\mathcal{D}$ be a dictionary satisfying Assumption 3.1. Let $W_K, W_Q \in \mathbb{R}^{d_h \times d}$ denote the parameters of an attention. Then for all $W_K, W_Q$ excluding a zero Lebesgue measure set in $\mathbb{R}^{d_h \times d} \times \mathbb{R}^{d_h \times d}$ and any $y_1, y_2, y_3 \in \mathcal{D}$, when

$$y_1 \neq y_2,$$

we have

$$y_1^\top W_K^\top W_Q y_3 \neq y_2^\top W_K^\top W_Q y_3.$$

*Proof.* Let $W_K$ be any full-rank (in the sense of row rank (if $d_h \leq d$) or column rank (if $d_h > d$)) matrix, because $0_d \notin \mathcal{D}$, we have

$$\mathcal{D} \cap \mathrm{Ker}(W_K) = \emptyset.$$

Here $\mathrm{Ker}$ is the kernel of the linear transformation whose matrix representation is $W_K$.

Define

$$\mathcal{Y} = \{y_1 - y_2 | y_1, y_2 \in \mathcal{D}, y_1 \neq y_2\} \cup \mathcal{D}$$

and

$$W_{y_3, y} = \{W | \mathrm{Sum}(W \odot W_K yy_3^\top) = 0, W \in \mathbb{R}^{d_h \times d}\}, \quad y_3 \in \mathcal{D}, \ y \in \mathcal{Y}.$$

Note that $0_d \notin \mathcal{Y}$ and $\mathcal{Y}$ is a finite set.

Hence we have

$$W_K y \neq 0_{d_h}, \quad y \in \mathcal{Y}.$$

Since $y_3 \neq 0_{d_h}$, $x \neq 0_d$, $W_{y_3, y}$ is a hyperplane in $\mathbb{R}^{d_h \times d}$.

This means

$$\mu_0(W_{y_3,y}) = 0,$$

where $\mu_0$ is the Lebesgue measure in $\mathbb{R}^{d_h \times d}$.

Define

$$W^* := \cup_{y_3 \in \mathcal{D}, x \in \mathcal{Y}} W_{y_3,y}$$

to be the union of all $W_{y_3,y}$.

We have

$$
\begin{aligned}
\mu_0(W^*) &= \mu_0(\cup_{y_3 \in \mathcal{D}, x \in \mathcal{Y}} W_{y_3,y}) \\
&\leq \sum_{y_3 \in \mathcal{D}, x \in \mathcal{Y}} \mu_0(W_{y_3,y}) \\
&= \sum_{y_3 \in \mathcal{D}, x \in \mathcal{Y}} 0 \\
&= 0
\end{aligned}
$$

This means for all $W_Q \in \mathbb{R}^{d_h \times d} / W^*$, where $W^*$ is a zero measure set, we have

$$\mathrm{Sum}(W_Q \odot W_K y y_3^\top) \neq 0, \quad y_3 \in \mathcal{D}, \ y \in \mathcal{Y}.$$

By Lemma A.1, this is equivalent to

$$(W_K y)^\top W_Q y_3 = \mathrm{Sum}(W_Q \odot (W_K y) y_3^\top) \neq 0.$$

Thus for any $y_1, y_2 \in \mathcal{D}$, set $y = y_1 - y_2$, we have

$$(W_K y_1 - W_K y_2)^\top W_Q y_3 \neq 0.$$

Furthermore, the feasible $W_K$ and $W_Q$ is selected from

$$(\mathbb{R}^{d_h \times d} / W^{**}) \times (\mathbb{R}^{d_h \times d} / W^*),$$

where $W^{**}$ denotes the set of all $d_h \times d$ matrices that aren't full-rank. Thus $W^{**}$ is a set with zero Lebesgue measure in $\mathbb{R}^{d_h \times d}$. This means that

$$
\begin{aligned}
\mu(\underbrace{\mathbb{R}^{d_h \times d} \times \mathbb{R}^{d_h \times d} / ((\mathbb{R}^{d_h \times d} / W^{**}) \times (\mathbb{R}^{d_h \times d} / W^*))}_{\text{the set excluded from the selection of } W_K, W_Q}) &= \mu((W^* \cup W^{**}) \times \mathbb{R}^{d_h \times d}) \\
&\leq \mu(W^* \times \mathbb{R}^{d_h \times d}) + \mu(W^{**} \times \mathbb{R}^{d_h \times d}) \\
&= 0 + 0 \\
&= 0
\end{aligned}
$$

where $\mu$ is the Lebesgue measure in $\mathbb{R}^{d_h \times d} \times \mathbb{R}^{d_h \times d}$.

This completes the proof.

$\square$

**Corollary A.0.1** (Attention with Positional Encoding Preserves Identity). Let $\mathcal{D}$ be a dictionary satisfying Assumption 3.1. Let $W_K \in \mathbb{R}^{d_h \times (d+n)}$ be of the form

$$W_K := [W_K^* \quad 0_{d_h \times n}] \in \mathbb{R}^{d_h \times (d+n)}.$$

Let $W_Q$ be a $d_h \times (d+n)$ matrix.
Let

$$R := [r_1, r_2, \cdots, r_n] \in \mathbb{R}^{n \times n}$$

be any positional encoding.

Let $y_1, y_2, y_3$ be any 3 tokens in $\mathcal{D}$ and let $y_1^p, y_2^p, y_3^p$ be any positional encoded version of these 3 tokens.

Then for all $W_K^*, W_Q$ excluding a zero Lebesgue measure set in $\mathbb{R}^{d_h \times d} \times \mathbb{R}^{d_h \times (d+n)}$, when

$$y_1 \neq y_2,$$

we have

$$(y_1^p)^\top W_K^\top W_Q y_3^p \neq (y_2^p)^\top W_K^\top W_Q y_3^p.$$

*Proof.* By the definition of $W_K$,

$$W_K \begin{bmatrix} x \\ r_i \end{bmatrix} = [W_K^* \quad 0_{d_h \times n}] \begin{bmatrix} x \\ r_i \end{bmatrix} \qquad \text{(Here } r_i \text{ is any column in } R)$$
$$= W_K^* x, \quad i \in [n].$$

Thus

$$W_K y_i^p = W_K^* y_i, \quad i \in \{1, 2\}.$$

Let $W_K^*$ be any full-rank matrix. Then $W_K^*(y_1 - y_2) \neq 0_{d_h}$ for any $y_1 \neq y_2$.

Thus

$$W_K^*(y_1 - y_2)(y_3^p)^\top \neq 0_{d_h \times (d+n)} \tag{A.1}$$

for any $y_1, y_2, y_3 \in \mathcal{D}$ and any positional encoding $r_i$ for $y_3$.

Define

$$\mathcal{D}^p := \{y + r_i | y \in \mathcal{D}, i \in [n]\}$$

be the set of all versions of positional encoded tokens in $\mathcal{D}$.

Define

$$\mathcal{Y} = \{y_1 - y_2 | y_1, y_2 \in \mathcal{D}, y_1 \neq y_2\} \cup \mathcal{D}$$

and

$$W_{y_3^p, y} = \{W | \text{Sum}(W \odot W_K y(y_3^p)^\top) = 0, W \in \mathbb{R}^{d_h \times (d+n)}\}, \quad y_3^p \in \mathcal{D}^p, \ y \in \mathcal{Y}.$$

Note that $0_d \notin \mathcal{Y}$ and $\mathcal{Y}$ is a finite set.

Since (A.1), $W_{y_3, y}$ is a hyperplane in $\mathbb{R}^{d_h \times (d+n)}$.

This means

$$\mu_0(W_{y_3^p, y}) = 0,$$

where $\mu_0$ is the Lebesgue measure in $\mathbb{R}^{d_h \times (d+n)}$.

Define

$$W^* := \cup_{y_3^p \in \mathcal{D}^p, y \in \mathcal{Y}} W_{y_3^p, y}$$

to be the union of all $W_{y_3^p, y}$.

We have

$$\mu_0(W^*) = \mu_0(\cup_{y_3^p \in \mathcal{D}^p, y \in \mathcal{Y}} W_{y_3^p, y})$$
$$\leq \sum_{y_3^p \in \mathcal{D}^p, y \in \mathcal{Y}} \mu_0(W_{y_3^p, y})$$
$$= \sum_{y_3^p \in \mathcal{D}^p, y \in \mathcal{Y}} 0$$

$$= 0$$

This means for all $W_Q \in \mathbb{R}^{d_h \times (d+n)}/W^*$, where $W^*$ is a zero measure set, we have

$$\text{Sum}(W_Q \odot W_K^* y (y_3^p)^\top) \neq 0, \quad y_3^p \in \mathcal{D}^p, \ y \in \mathcal{Y}.$$

By Lemma A.1, this is equivalent to

$$(W_K^* y)^\top W_Q y_3^p = \text{Sum}(W_Q \odot (W_K^* y)(y_3^p)^\top) \neq 0.$$

Thus for any $y_1, y_2 \in \mathcal{D}$, set $y = y_1 - y_2$, we have

$$(W_K^* y_1 - W_K^* y_2)^\top W_Q y_3^p \neq 0.$$

Furthermore, the feasible $W_K^*$ and $W_Q$ is selected from

$$(\mathbb{R}^{d_h \times d}/W^{**}) \times (\mathbb{R}^{d_h \times (d+n)}/W^*),$$

where $W^{**}$ denotes the set of all $d_h \times (d+n)$ matrices that aren't full-rank. Thus $W^{**}$ is a set with zero Lebesgue measure in $\mathbb{R}^{d_h \times (d+n)}$. This means that

$$\overbrace{\mu(\mathbb{R}^{d_h \times d} \times \mathbb{R}^{d_h \times (d+n)}/((\mathbb{R}^{d_h \times d}/W^{**}) \times (\mathbb{R}^{d_h \times (d+n)}/W^*)))}^{\text{the set excluded from the selection of } W_K^*, W_Q}$$

$$= \mu(W^{**} \times \mathbb{R}^{d_h \times (d+n)} \cup \mathbb{R}^{d_h \times d} \times W^*)$$

$$\leq \mu(W^{**} \times \mathbb{R}^{d_h \times (d+n)}) + \mu(\mathbb{R}^{d_h \times d} \times W^*)$$

$$= 0 + 0$$

$$= 0$$

where $\mu$ is the Lebesgue measure in $\mathbb{R}^{d_h \times (d+n)} \times \mathbb{R}^{d_h \times (d+n)}$.

This completes the proof.

$\square$

**Lemma A.3** (Least Separation). Let $\mathcal{D}$ be a dictionary satisfying Assumption 3.1. Let $X$ be an $n$-length input sequence

$$X := \begin{bmatrix} x_1 & x_2 & \cdots & x_n \end{bmatrix}, \quad x_i \in \mathcal{D}, i \in [n].$$

Let $R \in \mathbb{R}^{n \times n}$ denote any given positional encoding matrix. Let each column in the positional encoded input sequence be denoted as

$$\begin{bmatrix} x_1^p & x_2^p & \cdots & x_n^p \end{bmatrix} := \begin{bmatrix} X \\ R \end{bmatrix}.$$

Select $W_K, W_Q$ as that in Corollary A.0.1. This guarantees that for any $y_1, y_2, y_3 \in \mathcal{D}$ and any positional encoding of them (noted as $y_1^p, y_2^p, y_3^p$), when $y_1 \neq y_2$

$$(y_1^p)^\top W_K^\top W_Q y_3^p \neq (y_2^p)^\top W_K^\top W_Q y_3^p.$$

Then there exists a $\delta_m > 0$ such that

$$\left| \frac{\exp(x_{i_2}^\top W_K^\top W_Q x_{i_1})}{\sum_{i=1}^n \exp(x_i^\top W_K^\top W_Q x_{i_1})} - \frac{\exp(x_{i_3}^\top W_K^\top W_Q x_{i_1})}{\sum_{i=1}^n \exp(x_i^\top W_K^\top W_Q x_{i_1})} \right| \geq \delta_m, \quad , i_1, i_2, i_3 \in [n], x_{i_2} \neq x_{i_3},$$

for any $x_1, x_2, \cdots, x_n \in \mathcal{D}$.

*Proof.* Define

$$\mathcal{D}^p := \{y + r_i | y \in \mathcal{D}, i \in [n]\}$$

Define

$$\delta'_m := \min_{y_1^p, y_2^p, y_3^p \in \mathcal{D}^p, y_1 \neq y_2} (|(y_1^p)^\top W_K^\top W_Q(y_3^p) - (y_2^p)^\top W_K^\top W_Q(y_3^p)|).$$

By the set of all $(y_1^p, y_2^p, y_3^p), y_1^p, y_2^p, y_3^p \in \mathcal{D}^p$ is finite (because $\mathcal{D}$ by Assumption 3.1 is finite), the set of all $|(y_1^p)^\top W_K^\top W_Q(y_3^p) - (y_2^p)^\top W_K^\top W_Q(y_3^p)|$ is finite. This means the minimum in the above equation exists.

By that

$$(y_1^p)^\top W_K^\top W_Q(y_3^p) - (y_2^p)^\top W_K^\top W_Q(y_3^p)$$

if and only if

$$x_1 = x_2,$$

we have

$$\delta'_m \neq 0.$$

Combining with $\delta'_m \geq 0$, we have

$$\delta'_m > 0$$

Define

$$M_{\text{SA}} := \min_{x_i^p \in \mathcal{D}^p, i \in [n], i_1, i_2 \in [n]} \left( \frac{\exp\left((x_{i_2}^p)^\top W_K W_Q x_{i_1}^p\right)}{\sum_{i=1}^n \exp\left((x_i^p)^\top W_K W_Q(x_{i_1}^p)\right)} \right)$$

By the trait of exponential functions, $M_{\text{SA}} > 0$.

Define

$$\delta_m := M_{\text{SA}} \cdot \delta'_m.$$

Now we prove the claim in the main text.

Without loss of generality, suppose $(x_{i_2}^p)^\top W_K^\top W_Q(x_{i_1}^p) > (x_{i_3}^p)^\top W_K^\top W_Q(x_{i_1}^p)$, we have

$$\left| \frac{\exp\left((x_{i_2}^p)^\top W_K^\top W_Q x_{i_1}^p\right)}{\sum_{i=1}^n \exp\left((x_i^p)^\top W_K^\top W_Q x_{i_1}^p\right)} - \frac{\exp\left((x_{i_3}^p)^\top W_K^\top W_Q x_{i_1}^p\right)}{\sum_{i=1}^n \exp\left((x_i^p)^\top W_K^\top W_Q x_{i_1}^p\right)} \right|$$

$$= \frac{\exp\left((x_{i_3}^p)^\top W_K^\top W_Q x_{i_1}^p\right)}{\sum_{i=1}^n \exp\left((x_i^p)^\top W_K^\top W_Q x_{i_1}^p\right)} |\exp\left((x_{i_2}^p - x_{i_3}^p)^\top W_K^\top W_Q x_{i_1}^p\right) - 1|$$

$$> \frac{\exp\left((x_{i_3}^p)^\top W_K^\top W_Q x_{i_1}^p\right)}{\sum_{i=1}^n \exp\left((x_i^p)^\top W_K^\top W_Q x_{i_1}^p\right)} (x_{i_2}^p - x_{i_3}^p)^\top W_K^\top W_Q x_{i_1}^p \qquad (\text{By } e^x > x + 1 \text{ for } x > 0)$$

$$\geq M_{\text{SA}} \cdot \delta'_m$$

$$= \delta_m.$$

This completes the proof.

$\square$

**Theorem A.1** (Restate of Theorem 3.1:Transformer Recognize Identity). For a Dictionary $\mathcal{D}$ satisfying Assumption 3.1, let

$$X := [x_1 \quad x_2 \quad \cdots \quad x_n], \quad x_i \in \mathcal{D}, i \in [n]$$

denote the input sequence. And let $X_p$ denote $X$ with any full-rank positional embedding $R \in \mathbb{R}^{n \times n}$

$$X_p := \begin{bmatrix} X \\ R \end{bmatrix}.$$

We also denote every column of it as

$$X_p = \begin{bmatrix} x_1^p & x_2^p & \cdots & x_n^p \end{bmatrix}.$$

Let $W_V \in \mathbb{R}^{(d+n)\times(d+n)}, W_K, W_Q \in \mathbb{R}^{d_h\times(d+n)}$ denote the parameters of a single-head attention SA, where $d_h \in \mathbb{N}_+$ is the number of the hidden dimension. We assume $d_h \geq d$ (hidden dimension is no less than the input dimension). Then there exists a one-layer feed-forward network FFN and a self-attention SA such that the output satisfies

$$\text{FFN} \circ \text{SA}(X_p) = \begin{bmatrix} \text{ID}(X) & \text{ID}(X) & \cdots & \text{ID}(X) \end{bmatrix}$$

*Proof Sketch.* We give a proof sketch of this theorem.

- Step 1. By choosing $W_K, W_Q$ according to Lemma A.2, we are able to assure that only the identical tokens has the same attention weights.

- Step 2. Use FFN to subtract all attention weights with the attention weight of the last token and filter out all the zero entries.

$\square$

*Proof.* Let $W_K^* \in \mathbb{R}^{d_h \times d}$ be **any full-rank matrix** (in the sense of row rank (if $d_h \leq d$) or column rank (if $d_h > d$)), and construct $W_K$ to be

$$W_K := [W_K^* \quad 0_{d_h \times n}] \in \mathbb{R}^{d_h \times (d+n)}$$

This means that

$$W_K X_p = [W_K^* \quad 0_{d_h \times n}] \cdot \begin{bmatrix} X \\ R \end{bmatrix}$$
$$= W_K^* X,$$

which also writes out as

$$W_K x_i^p = W_K^* x_i, \quad i \in [n].$$

Then select $W_Q$ according to Corollary A.0.1, this assures that

$$(y_1^p)^\top W_K^\top W_Q y_3^p \neq (y_2^p)^\top W_K^\top W_Q y_3^p$$

if and only if

$$y_1 = y_2,$$

for every $y_1, y_2, y_3 \in \mathcal{D}$ and any positional encoding by $R$.

We also note that $W_Q$ can be **almost any matrix** in $\mathbb{R}^{d_h \times (d+n)}$ (excluding a zero measure set) according to Lemma A.2.

Let $W_V$ be **any full-rank matrix** in $\mathbb{R}^{(d+n)\times(d+n)}$.

According to Lemma A.3, there exists a $\delta_m$ such that for any $x_1, x_2, \cdots, x_n \in \mathcal{D}$ and any $i_1, i_2, i_3 \in [n]$, when $x_{i_2} \neq x_{i_3}$, we have

$$\left| \frac{\exp\left((x_{i_2}^p)^\top W_K^\top W_Q x_{i_1}^p\right)}{\sum_{i=1}^n \exp\left((x_i^p)^\top W_K^\top W_Q x_{i_1}^p\right)} - \frac{\exp\left((x_{i_3}^p)^\top W_K^\top W_Q x_{i_1}^p\right)}{\sum_{i=1}^n \exp\left((x_i^p)^\top W_K^\top W_Q x_{i_1}^p\right)} \right| \geq \delta_m. \tag{A.2}$$

Now we construct FFN : $\mathbb{R}^{d+n} \to \mathbb{R}^n$ according to the constructed attention.

$$\text{FFN}(Z) = \frac{2}{\delta_m}[\text{ReLU}(Z^* + v_\delta) - \text{ReLU}(Z^* + \frac{v_\delta}{2})] + \frac{2}{\delta_m}[\text{ReLU}(-Z^* + v_\delta) - \text{ReLU}(-Z^* + \frac{v_\delta}{2})] - 1_{n \times n},$$
$$\tag{A.3}$$

in which

$$Z^* := \begin{bmatrix} 0_{n \times d} & ER^{-1} \end{bmatrix} W_V^{-1} Z,$$

$$v_\delta := \delta_m 1_n,$$

$$E := \begin{bmatrix} 0_{1 \times (n-1)} & 0 \\ I_{n-1} & -1_{n-1} \end{bmatrix}.$$

The FFN we constructed transforms $Z$ to $Z^*$ and detect the entries in $Z^*$ that are exactly $0$.

We note that when

$$Z = W_V X_p \operatorname{Softmax}((W_K X_p)^\top W_Q X_p), \tag{A.4}$$

which is the output of SA, we have

$$
\begin{aligned}
Z^* &= \begin{bmatrix} 0_{n \times d} & ER^{-1} \end{bmatrix} W_V^{-1} \cdot W_V X_p \operatorname{Softmax}((W_K X_p)^\top W_Q X_p) \\
&= \begin{bmatrix} 0_{n \times d} & ER^{-1} \end{bmatrix} \begin{bmatrix} X \\ R \end{bmatrix} \operatorname{Softmax}((W_K X_p)^\top W_Q X_p) \\
&= E \operatorname{Softmax}((W_K X_p)^\top W_Q X_p) \\
&= \underbrace{\begin{bmatrix} 0_{1 \times (n-1)} & 0 \\ I_{n-1} & -1_{n-1} \end{bmatrix}}_{E} \operatorname{Softmax}((W_K X_p)^\top W_Q X_p).
\end{aligned}
$$

Because

$$
\begin{aligned}
&\begin{bmatrix} 0_{1 \times (n-1)} & 0 \\ I_{n-1} & -1_{n-1} \end{bmatrix} \operatorname{Softmax}((W_K X_p)^\top W_Q X_p) \\
&= \begin{bmatrix} 0_{1 \times (n-1)} & 0 \\ I_{n-1} & -1_{n-1} \end{bmatrix} \underbrace{\begin{bmatrix} \operatorname{Softmax}((W_K X_p)^\top W_Q X_p)_{1:n-1,:} \\ \operatorname{Softmax}((W_K X_p)^\top W_Q X_p)_{n,:} \end{bmatrix}}_{\operatorname{Softmax}((W_K X_p)^\top W_Q X_p)} \\
&= \begin{bmatrix} 0_{1 \times (n-1)} \cdot \operatorname{Softmax}((W_K X_p)^\top W_Q X_p)_{1:n-1,:} + 0 \cdot \operatorname{Softmax}((W_K X_p)^\top W_Q X_p)_{n,:} \\ I_{n-1} \cdot \operatorname{Softmax}((W_K X_p)^\top W_Q X_p)_{1:n-1,:} - 1_{n-1} \operatorname{Softmax}((W_K X_p)^\top W_Q X_p)_{n,:} \end{bmatrix} \\
&= \begin{bmatrix} 0_{1 \times n} \\ \operatorname{Softmax}((W_K X_p)^\top W_Q X_p)_{1:n-1,:} - 1_{n-1} \operatorname{Softmax}((W_K X_p)^\top W_Q X_p)_{n,:} \end{bmatrix}
\end{aligned}
$$

We have

$$Z^* = \begin{bmatrix} 0_{1 \times n} \\ \operatorname{Softmax}((W_K X_p)^\top W_Q X_p)_{1:n-1,:} - 1_{n-1} \operatorname{Softmax}((W_K X_p)^\top W_Q X_p)_{n,:} \end{bmatrix}.$$

For $r \in [n]/\{1\}$,

$$
\begin{aligned}
Z^*_{r,:} &= \begin{bmatrix} 0_{1 \times n} \\ \operatorname{Softmax}((W_K X_p)^\top W_Q X_p)_{1:n-1,:} - 1_{n-1} \operatorname{Softmax}((W_K X_p)^\top W_Q X_p)_{n,:} \end{bmatrix}_{r,:} \\
&= [\underbrace{\operatorname{Softmax}((W_K X_p)^\top W_Q X_p)_{1:n-1,:}}_{\text{first } n-1 \text{ row of } \operatorname{Softmax}((W_K X_p)^\top W_Q X_p)} - \underbrace{1_{n-1} \operatorname{Softmax}((W_K X_p)^\top W_Q X_p)_{n,:}}_{\text{every row is the same}}]_{r-1,:} \\
&= \operatorname{Softmax}((W_K X_p)^\top W_Q X_p)_{r-1,:} - \operatorname{Softmax}((W_K X_p)^\top W_Q X_p)_{n,:} \\
&= \frac{\overbrace{\exp\big((W_K^* x_r)^\top W_Q X_p\big)}^{1 \times n} - \overbrace{\exp\big((W_K^* x_n)^\top W_Q X_p\big)}^{1 \times n}}{\underbrace{\sum_{r'=1}^{n} \exp\big((W_K^* x_{r'})^\top W_Q X_p\big)}_{1 \times n}}. \qquad \text{\small (The division here is entry-wise)}
\end{aligned}
$$

This yields each entry to be

$$|Z^*_{r,c}| = \big|\frac{\exp\big((W^*_K x_r)^\top W_Q X_p\big) - \exp\big((W^*_K x_n)^\top W_Q X_p\big)}{\sum_{r'=1}^n \exp((W^*_K x_{r'})^\top W_Q X_p)}\big|_{:,c}$$

$$= \big|\frac{\exp\big((W^*_K x_r)^\top W_Q x^p_c\big) - \exp\big((W^*_K x_n)^\top W_Q x^p_c\big)}{\sum_{r'=1}^n \exp((W^*_K x_{r'})^\top W_Q x^p_c)}\big|, \quad c \in [n].$$

Combining the above equation with (A.2), we have

$$\begin{cases} |Z^*_{r,c}| \ge \delta_m, \ x_c \ne x_n \\ Z^*_{r,c} = 0, \ x_c = x_n \end{cases} \tag{A.5}$$

From the definition of FFN ((A.3)), we have

$$\mathrm{FFN}(Z)_{r,c} = \frac{2}{\delta_m}[\mathrm{ReLU}(Z^* + v_\delta)_{r,c} - \mathrm{ReLU}(Z^* + \frac{v_\delta}{2})_{r,c}] + \frac{2}{\delta_m}[\mathrm{ReLU}(-Z^* + v_\delta)_{r,c} - \mathrm{ReLU}(Z^* + \frac{v_\delta}{2})_{r,c}] - 1$$

$$= \frac{2}{\delta_m}[\mathrm{ReLU}(Z^*_{r,c} + \delta_m) - \mathrm{ReLU}(Z^*_{r,c} + \frac{\delta_m}{2})] + \frac{2}{\delta_m}[\mathrm{ReLU}(-Z^*_{r,c} + \delta_m) - \mathrm{ReLU}(-Z^*_{r,c} + \frac{\delta_m}{2})] - 1$$

For the two occasions of $Z^*_{r,c}$ as described in (A.5), when

$$Z^*_{r,c} = 0, \qquad\qquad\qquad \text{(Equivalent to } x_c = x_n)$$

we have

$$\mathrm{FFN}(Z)_{r,c} = \frac{2}{\delta_m}[\mathrm{ReLU}(\delta_m) - \mathrm{ReLU}(\frac{\delta_m}{2})] + \frac{2}{\delta_m}[\mathrm{ReLU}(\delta_m) - \mathrm{ReLU}(\frac{\delta_m}{2})] - 1$$

$$= (\frac{2}{\delta_m} \cdot \frac{\delta_m}{2} + \frac{2}{\delta_m} \cdot \frac{\delta_m}{2}) - 1$$

$$= 1$$

When

$$|Z^*_{r,c}| \ge \delta_m,, \qquad\qquad\qquad \text{(Equivalent to } x_c \ne x_n)$$

without loss of generalization, assume $Z^*_{r,c} \ge \delta_m$. We have

$$\mathrm{FFN}(Z)_{r,c} = \frac{2}{\delta_m}[(Z^*_{r,c} + \delta_m) - (Z^*_{r,c} + \frac{\delta_m}{2})] + \frac{2}{\delta_m} \cdot 0 - 1$$

$$= 1 - 1$$

$$= 0.$$

In conclusion, we have

$$\mathrm{FFN}(Z)_{r,c} = \begin{cases} 0, \ x_c \ne x_n \\ 1, \ x_c = x_n \end{cases}.$$

This is equivalent to

$$\mathrm{FFN}(Z) = [\mathrm{ID}(X) \quad \mathrm{ID}(X) \quad \cdots \quad \mathrm{ID}(X)],$$

for $Z = W_V X_p \, \mathrm{Softmax}((W_K X_p)^\top W_Q X_p)$ as noted in (A.4).

This is equivalent to

$$\mathrm{FFN} \circ \mathrm{SA}(X_p) = [\mathrm{ID}(X) \quad \mathrm{ID}(X) \quad \cdots \quad \mathrm{ID}(X)].$$

This completes the proof.

$\square$

**Corollary A.1.1** (Restate of Corollary 3.1.1: Choice of Attention)**.** Define

$$W_K := [W_K^* \quad 0_{d_h \times n}],$$

where $W_K^*$ is any full-rank matrix (in the sense of column rank).
Let $W_Q$ be almost any matrix in $\mathbb{R}^{d_h \times (d+n)}$ (excluding a zero measure set). Let $W_V$ be any full rank matrix in $\mathbb{R}^{(d+n) \times (d+n)}$.
Then for an attention SA parameterized by $W_K, W_Q, W_V$, there exists an FFN such that

$$\text{FFN} \circ \text{SA}(X_p) = [\text{ID}(X) \quad \text{ID}(X) \quad \cdots \quad \text{ID}(X)]$$

*Proof.* This is straightly given by the proof of Theorem A.1. The ranges of the choice of $W_K, W_Q, W_V$ are given in bold letters. $\square$

### A.2 Proofs of Results in Section 3.2

We first restate the definition of the induction head we use in this paper.

**Definition A.3** (Induction Head)**.** Let $X$ denote the input sequence and $x_i$, $i \in [n]$ denote its tokens, written as

$$X := [x_1 \quad x_2 \quad \cdots \quad x_n].$$

Let $D_n$ denote all labels of the tokens in $X$ that are identical to $x_n$, written as

$$D_n := \{i \mid x_i = x_n, i \neq n\}.$$

When there's only one token identical to $x_n$ in the whole sequence (except $x_n$), we define $\text{IND}(X)$ as the token next to that token.
When there are multiple tokens identical to $x_n$, we define the induction head $\text{IND}(X)$ as the average of the tokens next to the ones in $D_n$, which is

$$\text{IND}(X) := \frac{1}{|D_n|} \sum_{i \in D_n} x_{i+1},$$

where $|D_n|$ is the cardinality of $D_n$.

**Remark A.1.** When there's no token identical to $x_n$, $\text{IND}(X)$ does not function and it becomes meaningless to discuss approximating it with a transformer. In this sense, we only consider approximating an induction head on the input domains where it functions (i.e., inputs with tokens identical to the last one). Thus, we omit this situation.

We now prove Theorem 3.2

**Theorem A.2** (Restate of Theorem 3.2:Two-Layer Single-Head Attention Applies Induction Head)**.** Let $\mathcal{D}$ denote a dictionary satisfying Assumption 3.1. Define

$$X := [x_1 \quad x_2 \quad \cdots \quad x_n],$$

and let $X_p$ denote $X$ with any full-rank positional encoding $R \in \mathbb{R}^{n \times n}$

$$X_p = \begin{bmatrix} X \\ R \end{bmatrix}.$$

Define

$$W_K' := [W_K^* \quad 0_{d_h \times n}],$$

where $W_K^*$ is **any full-rank** (column rank) matrix. $d_h$ is the hidden dimension. We assume $d_h \geq d$ (hidden dimension is no less than the input dimension).
Let $W_Q'$ be **almost any matrix** in $\mathbb{R}^{d_h \times (d+n)}$ (excluding a zero measure set). Let $W_V'$ be **any full rank matrix** in $\mathbb{R}^{(d+n) \times (d+n)}$. We assume the hidden dimension to be no less than the input dimension ($d_h \geq d$).

Let $SA_1$ be a single-head self attention parametrized by $W_K', W_Q', W_V'$. Let $A_c \in \mathbb{R}^n$ be any vector whose entries are positive and sums up to be $1$. Then there exists an FFN and a single-head self attention $SA_2$ such that when the transformer comprised of $FFN \circ SA_1$ has skip connection, we have

$$\|SA_2(FFN \circ SA_1(X_p) + X_p) \cdot A_c - IND(X)\|_\infty \le \epsilon,$$

for any $\epsilon > 0$ when at least one token $x_i, i \ne n$ is identical to $x_n$.

**Remark A.2** (The Practical Necessity of $A_c$). In the context of autoregressive generative models (where the concept of induction head originates), the model takes a whole sequence and outputs a token.

Because attention and the token-wise feed-forward network don't change the sequence length, it is necessary to have an operation that transforms the processed sequence to an output token, which is usually the last layer.

In Theorem 3.2 this operation is denoted as $A_c \in \mathbb{R}^n$, which has a wide range of choices for applying an induction head.

*Proof.* By Corollary 3.1.1, because the condition of $W_K', W_Q', W_V'$ align with that of $W_K, W_Q, W_V$ in Corollary 3.1.1, there exists a self attention $SA_1$ parametrized by $W_K', W_Q', W_V'$ and a feed-forward layer $FFN^*$ such that

$$FFN^* \circ SA_1(X_p) = [ID(X) \quad ID(X) \quad \cdots \quad ID(X)]. \tag{A.6}$$

Construct FFN on the base of $FFN^*$ to be

$$FFN(Z) := \underbrace{\begin{bmatrix} 0_{d \times n} \\ m(R^\top)^{-1} \end{bmatrix}}_{(d+n) \times n} \cdot FFN^*(Z).$$

Here $m \in R$ is a coefficient that we configure in the later discussions.

Combine the above equation with (A.6) yields

$$FFN \circ SA(X_p) = \begin{bmatrix} 0_{d \times n} \\ m(R^\top)^{-1} \end{bmatrix} \cdot \underbrace{[ID(X) \quad ID(X) \quad \cdots \quad ID(X)]}_{ID(X) \cdot 1_{1 \times n}}$$

$$= \underbrace{\begin{bmatrix} 0_{d \times n} \\ m(R^\top)^{-1} ID(X) \cdot 1_{1 \times n} \end{bmatrix}}_{(d+n) \times n} \tag{A.7}$$

Along with skip connection, the input to the second attention layer is

$$FFN \circ SA(X_p) + X_p = \begin{bmatrix} 0_{d \times n} \\ m(R^\top (W_K^R)^\top W_Q^R)^{-1} ID(X) \cdot 1_{1 \times n} \end{bmatrix} + \begin{bmatrix} X \\ R \end{bmatrix}$$

$$= \begin{bmatrix} X \\ R + m(R^\top)^{-1} ID(X) \cdot 1_{1 \times n} \end{bmatrix} \tag{A.8}$$

**Calculation of the Output of** $SA_2$. For $SA_2$, we construct its $W_K, W_Q$ matrices to be

$$W_K := \begin{bmatrix} 0_{n \times d} & \frac{1}{m} I_n \end{bmatrix},$$
$$W_Q := \begin{bmatrix} 0_{n \times d} & \sqrt{m} I_n \end{bmatrix}.$$

Since (A.8) gives us the input to $SA_2$, the $K$ and $Q$ matrices of $SA_2$ are

$$K := W_K(FFN \circ SA(X_p) + X_p)$$

$$= \begin{bmatrix} 0_{n \times d} & \frac{1}{m} I_n \end{bmatrix} \cdot \begin{bmatrix} X \\ R + m(R^\top)^{-1} ID(X) \cdot 1_{1 \times n} \end{bmatrix} \quad \text{(By the definition of } W_K \text{ in the main text)}$$

$$= 0_{n \times d} \cdot X + \frac{1}{m} I_n \cdot (R + m(R^\top)^{-1} \mathrm{ID}(X) \cdot 1_{1 \times n})$$

$$= \frac{1}{m} R + (R^\top)^{-1} \mathrm{ID}(X) \cdot 1_{1 \times n}$$

and

$$Q := W_Q (\mathrm{FFN} \circ \mathrm{SA}(X_p) + X_p)$$

$$= \begin{bmatrix} 0_{n \times d} & \sqrt{m} I_n \end{bmatrix} \cdot \begin{bmatrix} X \\ R + m(R^\top)^{-1} \mathrm{ID}(X) \cdot 1_{1 \times n} \end{bmatrix}$$

$$= 0_{n \times d} \cdot X + \sqrt{m} I_n \cdot (R + m(R^\top)^{-1} \mathrm{ID}(X) \cdot 1_{1 \times n})$$

$$= \sqrt{m}(R + m(R^\top)^{-1} \mathrm{ID}(X) \cdot 1_{1 \times n})$$

The above two equations yield $K^\top Q$ to be

$$K^\top Q = [\frac{1}{m} R + (R^\top)^{-1} \mathrm{ID}(X) \cdot 1_{1 \times n}]^\top \cdot \sqrt{m}(R + m(R^\top)^{-1} \mathrm{ID}(X) \cdot 1_{1 \times n})$$

$$= \frac{1}{\sqrt{m}} R^\top R + \frac{1}{\sqrt{m}} R^\top \cdot m(R^\top)^{-1} \mathrm{ID}(X) \cdot 1_{1 \times n} + [(R^\top)^{-1} \mathrm{ID}(X) \cdot 1_{1 \times n}]^\top \cdot (R + m(R^\top)^{-1} \mathrm{ID}(X) \cdot 1_{1 \times n})$$

$$= \frac{1}{\sqrt{m}} R^\top R + \sqrt{m} \mathrm{ID}(X) \cdot 1_{1 \times n} + \underbrace{[\mathrm{ID}(X) \cdot 1_{1 \times n}]^\top}_{B^\top} \underbrace{[(R^\top)^{-1}]^\top}_{A^\top} \cdot (R + m(R^\top)^{-1} \mathrm{ID}(X) \cdot 1_{1 \times n})$$

$$\qquad\qquad\qquad\qquad\qquad\qquad\qquad\qquad\qquad\qquad ((AB)^\top = B^\top A^\top)$$

$$= \frac{1}{\sqrt{m}} R^\top R + \sqrt{m} \mathrm{ID}(X) \cdot 1_{1 \times n} + (\mathrm{ID}(X) \cdot 1_{1 \times n})^\top R^{-1} \cdot (R + m(R^\top)^{-1} \mathrm{ID}(X) \cdot 1_{1 \times n})$$

$$= \frac{1}{\sqrt{m}} R^\top R + \sqrt{m} \mathrm{ID}(X) \cdot 1_{1 \times n} + \underbrace{(\mathrm{ID}(X) \cdot 1_{1 \times n})^\top (I + m(RR^\top)^{-1} \mathrm{ID}(X) \cdot 1_{1 \times n})}_{\text{negligible in Softmax}} \cdot$$

For the last row in the above equations (especially the comment in it), we note that in $\mathrm{Softmax}(K^\top Q)$, because the $i$-th row in

$$(\mathrm{ID}(X) \cdot 1_{1 \times n})^\top (I + m(RR^\top)^{-1} \mathrm{ID}(X) \cdot 1_{1 \times n})$$

is

$$\underbrace{e_i^\top}_{\text{retreive the } i\text{-th row}} \cdot (\mathrm{ID}(X)^\top \cdot 1_{1 \times n})^\top (I + m(RR^\top)^{-1} \mathrm{ID}(X) \cdot 1_{1 \times n})$$

$$= \mathrm{ID}(X)(I + m(RR^\top)^{-1} \mathrm{ID}(X) \cdot 1_{1 \times n}),$$

which is invariant of $i$.

This means for any $j \in [n]$, the $j$-th entry of every row is the same.

Thus the columns of

$$(\mathrm{ID}(X) \cdot 1_{1 \times n})^\top (I + m(RR^\top)^{-1} \mathrm{ID}(X) \cdot 1_{1 \times n})$$

are columns with identical value on every row.

Because identical values in the columns is negligible to $\mathrm{Softmax}$, we have the comment in the original equation. This formally writes out as

$$\mathrm{Softmax}(K^\top Q) = \mathrm{Softmax}(\frac{1}{\sqrt{m}} R^\top R + \sqrt{m} \mathrm{ID}(X) \cdot 1_{1 \times n} + (\mathrm{ID}(X) \cdot 1_{1 \times n})^\top (I + m(RR^\top)^{-1} \mathrm{ID}(X) \cdot 1_{1 \times n}))$$

$$= \mathrm{Softmax}(\frac{1}{\sqrt{m}} R^\top R + \sqrt{m} \mathrm{ID}(X) \cdot 1_{1 \times n})$$

Construct $W_V$ to be

$$W_V := \begin{bmatrix} I_d & 0_{d \times n} \end{bmatrix}$$

Since $W_V = I_{d+n}$, the $V$ matrix of $\text{SA}_2$ is

$$V := W_V(\text{FFN} \circ \text{SA}(X_p) + X_p)$$

$$= [I_d \quad 0_{d \times n}] \cdot \begin{bmatrix} X \\ R + m(R^\top)^{-1}\text{ID}(X) \cdot 1_{1 \times n} \end{bmatrix} \qquad \text{(By (A.8))}$$

$$= X$$

Thus the output of $\text{SA}_2$ is

$$\text{SA}_2(\text{FFN} \circ \text{SA}(X_p) + X_p) := V \, \text{Softmax}(K^\top Q)$$

$$= X \, \text{Softmax}(\frac{1}{\sqrt{m}} R^\top R + \sqrt{m}\text{ID}(X) \cdot 1_{1 \times n})$$

Denote $R^\top R$ as $T$ for simplicity. The $j$-th column of the output of $\text{SA}_2$ is

$$X \, \text{Softmax}(\frac{1}{\sqrt{m}} \underbrace{T}_{R^\top R} + \sqrt{m}\text{ID}(X) \cdot 1_{1 \times n})_{:,j} = [x_1 \quad x_2 \quad \cdots \quad x_n] \cdot \text{Softmax}(\frac{1}{\sqrt{m}} T_{:,j} + \sqrt{m}\text{ID}(X))$$

$$= \sum_{i=1}^n \frac{\exp\left(\frac{1}{\sqrt{m}} T_{i,j} + \sqrt{m}\text{ID}(X)_i\right)}{\sum_{i'=1}^n \exp\left(\frac{1}{\sqrt{m}} T_{i',j} + \sqrt{m}\text{ID}(X)_{i'}\right)} \cdot x_i.$$

$$(A.9)$$

Let's consider the coefficient before each $x_i$, which is

$$\frac{\exp\left(\frac{1}{\sqrt{m}} T_{i,j} + \sqrt{m}\text{ID}(X)_i\right)}{\sum_{i'=1}^n \exp\left(\frac{1}{\sqrt{m}} T_{i',j} + \sqrt{m}\text{ID}(X)_{i'}\right)}.$$

Because $\text{ID}(X)_i$ is either 1 or 0 by its definition in Definition A.1. We therefore divide the above coefficients into two cases

$$\frac{\exp\left(\frac{1}{\sqrt{m}} T_{i,j} + \sqrt{m}\text{ID}(X)_i\right)}{\sum_{i'=1}^n \exp\left(\frac{1}{\sqrt{m}} T_{i',j} + \sqrt{m}\text{ID}(X)_{i'}\right)}$$

$$= \begin{cases} \frac{\exp\left(\frac{1}{\sqrt{m}} T_{i,j} + \sqrt{m}\right)}{\sum_{i'=1}^n \exp\left(\frac{1}{\sqrt{m}} T_{i',j} + \sqrt{m}\text{ID}(X)_{i'}\right)}, & \text{ID}(X)_i = 1, \\ \frac{\exp\left(\frac{1}{\sqrt{m}} T_{i,j}\right)}{\sum_{i'=1}^n \exp\left(\frac{1}{\sqrt{m}} T_{i',j} + \sqrt{m}\text{ID}(X)_{i'}\right)}, & \text{ID}(X)_i = 0. \end{cases}$$

Because the denominators in the two cases are both

$$\sum_{i'=1}^n \exp\left(\frac{1}{\sqrt{m}} T_{i',j} + \sqrt{m}\text{ID}(X)_{i'}\right).$$

The ratio between a coefficient (labeled as $i_1$) whose $\text{ID}(X)_{i_1} = 1$ and another one (labeled as $i_2$) whose $\text{ID}(X)_{i_2} = 0$ is the ratio of their nominators, which is

$$\frac{\exp\left(\frac{1}{\sqrt{m}} T_{i_1,j} + \sqrt{m}\right)}{\exp\left(\frac{1}{\sqrt{m}} T_{i_1,j}\right)} = \exp\left(\frac{1}{\sqrt{m}}(T_{i_1,j} - T_{i_2,j}) + \sqrt{m}\right) \geq \exp\left(\sqrt{m} - \frac{2}{\sqrt{m}} B_T\right), \quad (A.10)$$

where $B_T := \|T\|_\infty$ is the maximal entry of $|T|$. Since $|T| = |R^\top R|$ ($R$ is the positional encoding), it has a maximal entry.

By Definition A.3, we only consider the situations where the induction head functions, i.e., when there's a token identical to the last token (except for the last toke itself). This means that at least one entry in $\text{ID}(X)$ is 1.

Define $M_0$ as

$$M_0 := \max_{\mathrm{ID}(X)_i=0} \left( \frac{\exp\left(\frac{1}{\sqrt{m}}T_{i,j}\right)}{\sum_{i'=1}^n \exp\left(\frac{1}{\sqrt{m}}T_{i',j} + \sqrt{m}\mathrm{ID}(X)_{i'}\right)} \right).$$

Since all

$$\frac{\exp\left(\frac{1}{\sqrt{m}}T_{i,j} + \sqrt{m}\mathrm{ID}(X)_i\right)}{\sum_{i'=1}^n \exp\left(\frac{1}{\sqrt{m}}T_{i',j} + \sqrt{m}\mathrm{ID}(X)_{i'}\right)}$$

sums up to be 1, we have

$$1 = \sum_{i=1}^n \frac{\exp\left(\frac{1}{\sqrt{m}}T_{i,j} + \sqrt{m}\mathrm{ID}(X)_i\right)}{\sum_{i'=1}^n \exp\left(\frac{1}{\sqrt{m}}T_{i',j} + \sqrt{m}\mathrm{ID}(X)_{i'}\right)} \cdot x_i$$

$$\geq \max_{\mathrm{ID}(X)_i=0} \left( \frac{\exp\left(\frac{1}{\sqrt{m}}T_{i,j}\right)}{\sum_{i'=1}^n \exp\left(\frac{1}{\sqrt{m}}T_{i',j} + \sqrt{m}\mathrm{ID}(X)_{i'}\right)} \right) + \frac{\exp\left(\frac{1}{\sqrt{m}}T_{i,j} + \sqrt{m}\right)}{\sum_{i'=1}^n \exp\left(\frac{1}{\sqrt{m}}T_{i',j} + \sqrt{m}\mathrm{ID}(X)_{i'}\right)}$$

$$\geq M_0 + \exp\left(\sqrt{m} - \frac{2}{\sqrt{m}}B_T\right)M_0. \qquad \text{(By (A.10))}$$

This means that

$$M_0 \leq \frac{1}{\exp\left(\sqrt{m} - \frac{2}{\sqrt{m}}B_T\right) + 1}.$$

For any $\epsilon_1 > 0$, when

$$m \geq \max((-\ln(\epsilon_1) + 2B_T)^2, 1),$$

we have

$$M_0 \leq \frac{1}{\exp\left(\sqrt{(-\ln(\epsilon_1) + 2B_T)^2} - \frac{2}{\sqrt{1}}B_T\right) + 1}$$

$$= \frac{1}{\exp(-\ln(\epsilon_1) + 2B_T - 2B_T) + 1}$$

$$= \frac{1}{-\ln(\epsilon_1) + 1}$$

$$\leq \epsilon_1. \qquad \text{(A.11)}$$

From this last inequality, we prove any coefficient of $x_i$:

$$\frac{\exp\left(\frac{1}{\sqrt{m}}T_{i,j} + \sqrt{m}\mathrm{ID}(X)_i\right)}{\sum_{i'=1}^n \exp\left(\frac{1}{\sqrt{m}}T_{i',j} + \sqrt{m}\mathrm{ID}(X)_{i'}\right)}$$

with $\mathrm{ID}(X)_i = 0$ is arbitrarily small when setting $m$ to be sufficiently large.

We now demonstrate the coefficients of $x_i$ which have $\mathrm{ID}(X)_i = 1$ are arbitrarily close to each other when $m$ is sufficiently large.

The ratio between any two coefficients of $x_{i_1}, x_{i_2}$ which satisfy $\mathrm{ID}(X)_{i_1} = 1, \mathrm{ID}(X)_{i_2} = 1$ is

$$\frac{\frac{\exp\left(\frac{1}{\sqrt{m}}T_{i_1,j} + \sqrt{m}\mathrm{ID}(X)_{i_1}\right)}{\sum_{i'=1}^n \exp\left(\frac{1}{\sqrt{m}}T_{i',j} + \sqrt{m}\mathrm{ID}(X)_{i'}\right)}}{\frac{\exp\left(\frac{1}{\sqrt{m}}T_{i_2,j} + \sqrt{m}\mathrm{ID}(X)_{i_2}\right)}{\sum_{i'=1}^n \exp\left(\frac{1}{\sqrt{m}}T_{i',j} + \sqrt{m}\mathrm{ID}(X)_{i'}\right)}} = \frac{\exp\left(\frac{1}{\sqrt{m}}T_{i_1,j} + \sqrt{m}\right)}{\exp\left(\frac{1}{\sqrt{m}}T_{i_2,j} + \sqrt{m}\right)}$$

$$= \exp\left(\frac{1}{\sqrt{m}}(T_{i_1,j} - T_{i_2,j})\right) \in [\exp\left(-\frac{2}{\sqrt{m}}B_T\right), \exp\left(\frac{2}{\sqrt{m}}B_T\right)]$$

For any $\epsilon_2 > 0$, when $m \geq (2B_T/\ln(\epsilon_2))^2$, by the above inequality we have

$$\frac{\frac{\exp\left(\frac{1}{\sqrt{m}}T_{i_1,j} + \sqrt{m}\text{ID}(X)_{i_1}\right)}{\sum_{i'=1}^{n}\exp\left(\frac{1}{\sqrt{m}}T_{i',j} + \sqrt{m}\text{ID}(X)_{i'}\right)}}{\frac{\exp\left(\frac{1}{\sqrt{m}}T_{i_2,j} + \sqrt{m}\text{ID}(X)_{i_2}\right)}{\sum_{i'=1}^{n}\exp\left(\frac{1}{\sqrt{m}}T_{i',j} + \sqrt{m}\text{ID}(X)_{i'}\right)}} \in [\exp\left(-\frac{2}{\sqrt{m}}B_T\right), \exp\left(\frac{2}{\sqrt{m}}B_T\right)]$$

$$\subset [\exp\left(-\frac{2}{\sqrt{(\frac{2B_T}{\ln(\epsilon_2)})^2}}B_T\right), \exp\left(\frac{2}{\sqrt{(\frac{2B_T}{\ln(\epsilon_2)})^2}}B_T\right)]$$

$$\subset [\exp(-\epsilon_2), \exp(\epsilon_2)] \tag{A.12}$$

Let $S$ denote the set of all $i$ satisfying $\text{ID}(X)_i = 1$, which is

$$S := \{i | \text{ID}(X)_i = 1\}.$$

Let $|S|$ denote the cardinality of $S$.

For any $k \in S$, plug (A.11) and (A.12) to

$$\sum_{i=1}^{n} \frac{\exp\left(\frac{1}{\sqrt{m}}T_{i,j} + \sqrt{m}\text{ID}(X)_i\right)}{\sum_{i'=1}^{n}\exp\left(\frac{1}{\sqrt{m}}T_{i',j} + \sqrt{m}\text{ID}(X)_{i'}\right)} = 1$$

yields

$$1 = \sum_{i=1}^{n} \frac{\exp\left(\frac{1}{\sqrt{m}}T_{i,j} + \sqrt{m}\text{ID}(X)_i\right)}{\sum_{i'=1}^{n}\exp\left(\frac{1}{\sqrt{m}}T_{i',j} + \sqrt{m}\text{ID}(X)_{i'}\right)}$$

$$= \underbrace{\sum_{\substack{\text{ID}(X)_{i_1}=1 \\ i_1 \in S}} \frac{\exp\left(\frac{1}{\sqrt{m}}T_{i_1,j} + \sqrt{m}\text{ID}(X)_{i_1}\right)}{\sum_{i'=1}^{n}\exp\left(\frac{1}{\sqrt{m}}T_{i',j} + \sqrt{m}\text{ID}(X)_{i'}\right)}} + \sum_{\text{ID}(X)_{i_2}=0} \frac{\exp\left(\frac{1}{\sqrt{m}}T_{i_2,j} + \sqrt{m}\text{ID}(X)_{i_2}\right)}{\sum_{i'=1}^{n}\exp\left(\frac{1}{\sqrt{m}}T_{i',j} + \sqrt{m}\text{ID}(X)_{i'}\right)}$$

$$\leq \underbrace{|S|\exp(\epsilon_2)\frac{\exp\left(\frac{1}{\sqrt{m}}T_{k,j} + \sqrt{m}\text{ID}(X)_k\right)}{\sum_{i'=1}^{n}\exp\left(\frac{1}{\sqrt{m}}T_{i',j} + \sqrt{m}\text{ID}(X)_{i'}\right)}}_{\text{corresponds to } i_1 \in S} + \underbrace{n\epsilon_1 \frac{\exp\left(\frac{1}{\sqrt{m}}T_{k,j} + \sqrt{m}\text{ID}(X)_k\right)}{\sum_{i'=1}^{n}\exp\left(\frac{1}{\sqrt{m}}T_{i',j} + \sqrt{m}\text{ID}(X)_{i'}\right)}}_{\text{corresponds to } i_2 \notin S}$$

$$\text{(By (A.11) and (A.12))}$$

The above inequality means that

$$\frac{\exp\left(\frac{1}{\sqrt{m}}T_{k,j} + \sqrt{m}\text{ID}(X)_k\right)}{\sum_{i'=1}^{n}\exp\left(\frac{1}{\sqrt{m}}T_{i',j} + \sqrt{m}\text{ID}(X)_{i'}\right)}(|S|\exp(\epsilon_2) + n\epsilon_1) \geq 1,$$

which is equivalent to

$$\frac{\exp\left(\frac{1}{\sqrt{m}}T_{k,j} + \sqrt{m}\text{ID}(X)_k\right)}{\sum_{i'=1}^{n}\exp\left(\frac{1}{\sqrt{m}}T_{i',j} + \sqrt{m}\text{ID}(X)_{i'}\right)} \geq \frac{1}{|S|\exp(\epsilon_2) + n\epsilon_1}. \tag{A.13}$$

Meanwhile we have

$$1 = \sum_{i=1}^{n} \frac{\exp\left(\frac{1}{\sqrt{m}}T_{i,j} + \sqrt{m}\text{ID}(X)_i\right)}{\sum_{i'=1}^{n}\exp\left(\frac{1}{\sqrt{m}}T_{i',j} + \sqrt{m}\text{ID}(X)_{i'}\right)}$$

$$= \sum_{\text{ID}(X)_{i_1}=1} \frac{\exp\left(\frac{1}{\sqrt{m}}T_{i_1,j} + \sqrt{m}\text{ID}(X)_{i_1}\right)}{\sum_{i'=1}^{n} \exp\left(\frac{1}{\sqrt{m}}T_{i',j} + \sqrt{m}\text{ID}(X)_{i'}\right)} + \sum_{\text{ID}(X)_{i_2}=0} \frac{\exp\left(\frac{1}{\sqrt{m}}T_{i_2,j} + \sqrt{m}\text{ID}(X)_{i_2}\right)}{\sum_{i'=1}^{n} \exp\left(\frac{1}{\sqrt{m}}T_{i',j} + \sqrt{m}\text{ID}(X)_{i'}\right)}$$

$$\geq \sum_{\text{ID}(X)_{i_1}=1} \frac{\exp\left(\frac{1}{\sqrt{m}}T_{i_1,j} + \sqrt{m}\text{ID}(X)_{i_1}\right)}{\sum_{i'=1}^{n} \exp\left(\frac{1}{\sqrt{m}}T_{i',j} + \sqrt{m}\text{ID}(X)_{i'}\right)} + 0$$

$$\geq |S| \exp(-\epsilon_2) \frac{\exp\left(\frac{1}{\sqrt{m}}T_{k,j} + \sqrt{m}\text{ID}(X)_k\right)}{\sum_{i'=1}^{n} \exp\left(\frac{1}{\sqrt{m}}T_{i',j} + \sqrt{m}\text{ID}(X)_{i'}\right)}. \qquad \text{(By (A.12))}$$

This means that

$$\frac{\exp\left(\frac{1}{\sqrt{m}}T_{k,j} + \sqrt{m}\text{ID}(X)_k\right)}{\sum_{i'=1}^{n} \exp\left(\frac{1}{\sqrt{m}}T_{i',j} + \sqrt{m}\text{ID}(X)_{i'}\right)} \leq \frac{1}{|S| \exp(-\epsilon_2)}$$

Combine this with (A.13) yields

$$\frac{1}{|S| \exp(\epsilon_2) + n\epsilon_1} \leq \frac{\exp\left(\frac{1}{\sqrt{m}}T_{k,j} + \sqrt{m}\text{ID}(X)_k\right)}{\sum_{i'=1}^{n} \exp\left(\frac{1}{\sqrt{m}}T_{i',j} + \sqrt{m}\text{ID}(X)_{i'}\right)} \leq \frac{1}{|S| \exp(-\epsilon_2)}.$$

This means when we set $\epsilon_2, \epsilon_1 > 0$ to be small enough.

We have for any $\epsilon_3 > 0$,

$$\left| \frac{\exp\left(\frac{1}{\sqrt{m}}T_{k,j} + \sqrt{m}\text{ID}(X)_k\right)}{\sum_{i'=1}^{n} \exp\left(\frac{1}{\sqrt{m}}T_{i',j} + \sqrt{m}\text{ID}(X)_{i'}\right)} - \frac{1}{|S|} \right| \leq \epsilon_3.$$

Combined with (A.9), we have

$$\|X \text{Softmax}(\frac{1}{\sqrt{m}} \underbrace{T}_{R^\top R} + \sqrt{m}\text{ID}(X) \cdot 1_{1 \times n})_{:,j} - \frac{1}{|S|} \sum_{i \in S} x_i\|_\infty$$

$$= \|(\sum_{i \in S} \frac{\exp\left(\frac{1}{\sqrt{m}}T_{i,j} + \sqrt{m}\text{ID}(X)_i\right)}{\sum_{i'=1}^{n} \exp\left(\frac{1}{\sqrt{m}}T_{i',j} + \sqrt{m}\text{ID}(X)_{i'}\right)} - \frac{1}{|S|}) \cdot x_i + \sum_{i \in [N]/S} \frac{\exp\left(\frac{1}{\sqrt{m}}T_{i,j} + \sqrt{m}\text{ID}(X)_i\right)}{\sum_{i'=1}^{n} \exp\left(\frac{1}{\sqrt{m}}T_{i',j} + \sqrt{m}\text{ID}(X)_{i'}\right)} \cdot x_i\|_\infty$$

$$\leq \|(\sum_{i \in S} \frac{\exp\left(\frac{1}{\sqrt{m}}T_{i,j} + \sqrt{m}\text{ID}(X)_i\right)}{\sum_{i'=1}^{n} \exp\left(\frac{1}{\sqrt{m}}T_{i',j} + \sqrt{m}\text{ID}(X)_{i'}\right)} - \frac{1}{|S|}) \cdot x_i\|_\infty + \|\sum_{i \in [n]/S} \frac{\exp\left(\frac{1}{\sqrt{m}}T_{i,j} + \sqrt{m}\text{ID}(X)_i\right)}{\sum_{i'=1}^{n} \exp\left(\frac{1}{\sqrt{m}}T_{i',j} + \sqrt{m}\text{ID}(X)_{i'}\right)} \cdot x_i\|_\infty$$

$$\leq |\sum_{i \in S} \frac{\exp\left(\frac{1}{\sqrt{m}}T_{i,j} + \sqrt{m}\text{ID}(X)_i\right)}{\sum_{i'=1}^{n} \exp\left(\frac{1}{\sqrt{m}}T_{i',j} + \sqrt{m}\text{ID}(X)_{i'}\right)} - \frac{1}{|S|}| \cdot \|x_i\|_\infty + \sum_{i \in [n]/S} \frac{\exp\left(\frac{1}{\sqrt{m}}T_{i,j} + \sqrt{m}\text{ID}(X)_i\right)}{\sum_{i'=1}^{n} \exp\left(\frac{1}{\sqrt{m}}T_{i',j} + \sqrt{m}\text{ID}(X)_{i'}\right)} \cdot \|x_i\|_\infty$$

$$\leq |S|\epsilon_3 B_x + (n - |S|)\epsilon_1 B_x.$$

Here $B_x$ denotes

$$\max_{x \in \mathcal{D}} (\|x\|_\infty).$$

Because $\mathcal{D}$ is finite, it exists.

Then set $\epsilon_3, \epsilon_1 \leq \epsilon/(B_x n)$. This yields

$$\|X \text{Softmax}(\frac{1}{\sqrt{m}} \underbrace{T}_{R^\top R} + \sqrt{m}\text{ID}(X) \cdot 1_{1 \times n})_{:,j} - \frac{1}{|S|} \sum_{i \in S} x_i\|_\infty \leq \epsilon$$

Because $i \in S$ is equivalent to $\mathrm{ID}(X)_i = 1$, which means $x_{i-1} = x_n$. The above inequality is equivalent to

$$\|X \mathrm{Softmax}(\frac{1}{\sqrt{m}} \underbrace{T}_{R^\top R} + \sqrt{m}\mathrm{ID}(X) \cdot 1_{1 \times n})_{:,j} - \frac{1}{|D_n|} \sum_{x_{i-1}=x_n} x_i\|_\infty \leq \epsilon,$$

which is

$$\|X \mathrm{Softmax}(\frac{1}{\sqrt{m}} \underbrace{T}_{R^\top R} + \sqrt{m}\mathrm{ID}(X) \cdot 1_{1 \times n})_{:,j} - \mathrm{IND}(X)\|_\infty \leq \epsilon$$

Because $A_c$ performs a weighted average over all columns, we have

$$\|X \mathrm{Softmax}(\frac{1}{\sqrt{m}} \underbrace{T}_{R^\top R} + \sqrt{m}\mathrm{ID}(X) \cdot 1_{1 \times n})A_c - \mathrm{IND}(X)\|_\infty \leq \epsilon(\sum_{i=1}^n (A_c)_i) = \epsilon 1_{1 \times n} \cdot A_c = \epsilon.$$

This completes the proof.

$\square$

### A.3 Proofs of Results in Section 3.3

**Theorem A.3** (Restate of Theorem 3.3:Multi-Head Attention Followed by Single-Head Attention Applies Induction Head). Let $X \in \mathbb{R}^{d \times n}$ denote the input sequence whose tokens are sampled from dictionary $\mathcal{D}$ satisfying Assumption 3.1. Let $X_p \in \mathbb{R}^{(d+n) \times n}$ denote the positional encoded version of $X$ and let $R \in \mathbb{R}^{n \times n}$ denote this positional encoding. Additionally, we assume $R$ to be a full-rank matrix.
Let MA be a multi-head attention and let $\mathrm{SA}_i, i \in [H]$ denote its $H$ heads. Let $W_K^i, W_Q^i \in \mathbb{R}^{(d+n)}$, $W_V^i$ denote the parameters of $\mathrm{SA}_i$.
If there exists an $i_0 \in [H]$, such that $W_K^{i_0} = [W_K^* \ 0_{d_h \times n}]$, $W_K^*$ is full-rank (in the sense of column rank), $W_Q^{i_0} \in \mathbb{R}^{d_h \times (d+n)}$ is not in a specific set of zero measure, and $W_V \in \mathbb{R}^{(d+n) \times (d+n)}$ is full-rank, then there exists a feed-forward network FFN and a self-attention SA such that

$$\|\mathrm{SA} \circ \mathrm{FFN} \circ \mathrm{MA}(X_p)A_c - \mathrm{IND}(X)\|_\infty \leq \epsilon,$$

for any $\epsilon > 0$. Here $A_c \in \mathbb{R}^n$ is any vector whose entries are positive and sum up to one.

*Proof.* Construct a linear transformation $A_1$ such that

$$A_1(Z) := Z_{(i_0-1)(d+n)+1:i_0(d+n),:},$$

or written in explicit construction as

$$A_1(Z) := \begin{bmatrix} 0_{(d+n) \times (i_0-1)(d+n)} & 0_{(d+n) \times (d+n)} & 0_{(d+n) \times (H-i_0)(d+n)} \\ 0_{(d+n) \times (i_0-1)(d+n)} & I_{d+n} & 0_{(d+n) \times (H-i_0)(d+n)} \\ 0_{(d+n) \times (i_0-1)(d+n)} & 0_{(d+n) \times (d+n)} & 0_{(d+n) \times (H-i_0)(d+n)} \end{bmatrix} Z$$

This yields

$$A_1 \circ \mathrm{MA}(X_p) = \mathrm{SA}_{i_0}(X_p),$$

By Theorem A.2, there exist two single-head self-attention layers connected by a feed-forward network that applies the induction head. Furthermore, the first layer of self-attention can be any attention that satisfies $W_K = [W_K^* \ 0_{d_h \times n}]$ where $W_K^*$ is full-rank (in the sense of column rank), $W_Q^{i_0} \in \mathbb{R}^{d_h \times (d+n)}$ isn't in a specific set of zero measure, and $W_V \in \mathbb{R}^{(d+n) \times (d+n)}$ is full-rank. Because $\mathrm{SA}_{i_0}$ fits these conditions, we choose $\mathrm{SA}_{i_0}$ as the attention in the first layer. Then by Theorem 3.2, there is a FFN and a $\mathrm{SA}_2$ and a $A_c \in \mathbb{R}^n$ whose entries are positive and sum up to 1, such that

$$\|\mathrm{SA} \circ \mathrm{FFN}_1 \circ \mathrm{SA}_{i_0}(X_p)A_c - \mathrm{IND}(X)\|_\infty \leq \epsilon,$$

which is equivalent to

$$\|\mathrm{SA} \circ \mathrm{FFN}_1 \circ A_1 \circ \mathrm{MA}(X_p)A_c - \mathrm{IND}(X)\|_\infty \le \epsilon,$$

Since $\mathrm{FFN} \circ A_1$ is still a feed-forward layer, we define it as FFN.

This completes the proof.

$\square$

### A.4 Proofs of Results in Section 3.4

**Lemma A.4** (Attention Maps Identical Tokens to Identical Tokens). For any self-attention SA and any input sequence $X$ for SA, if its two tokens $x_i, x_j$ satisfies $x_i = x_j$, then

$$\mathrm{SA}(X)_i = \mathrm{SA}(X)_j.$$

*Proof.* Let $W_K, W_Q, W_V$ denote the parameters of SA, then

$$\mathrm{SA}(X)_i := W_V X \left[(W_K X)^\top W_Q x_i\right] = W_V X \left[(W_K X)^\top W_Q x_j\right] = \mathrm{SA}(X)_j.$$

This completes the proof. $\square$

**Theorem A.4** (Restate of Theorem 3.4: Latent Space Induction Head of Multi-Layer Single-Head Attention Network). Let $X$ denote the input sequence satisfying Assumption 3.2 and let $X_p$ be the positional encoded version of it of positional encoding $I_n$. Let $F$ be any $N$-layer network of single-head attention whose $W_V$ matrices satisfy Assumption 3.3. Let $\theta_f$ denote the parameter of all $W_K$ and $W_Q$ matrices in $F$ and let $d_F$ denote its dimension.
For any $\delta, \epsilon > 0$, $N \in \mathbb{N}^+$ and any $\theta_F \in \mathbb{R}^{d_F}$ except for a subset in $\mathbb{R}^{d_f}$ of arbitrarily small measure, there exists a single-head attentions $\mathrm{SA}_1$ and two single-head transformers $T_1, T_2$ such that

$$\|T_2 \circ T_1(F \circ \mathrm{SA}_1(X_p) + X_p)A_c - \mathrm{LaIND}_{F \circ \mathrm{SA}_1 + I}(X_p)\|_\infty \le \epsilon,$$
$$\|\mathrm{SA}_1(X_p)_{1:d,:} - X\|_\infty \le \delta,$$

where $A_c \in \mathbb{R}^n$ is any vector whose entries are positive and sum up to be one. The second condition on $\mathrm{SA}_1$ means $\mathrm{SA}_1$ almost only changes the positional encoding.

*Proof.* Construct $\mathrm{SA}_1$ to be

$$\mathrm{SA}_1(Z) := Z \, \mathrm{Softmax}(T([I_d \quad 0_{d\times n}] Z)^\top [I_d \quad 0_{d\times n}] Z),$$

where $Z \in \mathbb{R}^{(d+n)\times n}$ denotes the input, and $T \in \mathbb{R}^+$ is a coefficient we use to control the precision of approximation in later discussion.

This yields $\mathrm{SA}_1(X_p)$ to be

$$\mathrm{SA}_1(X_p) = X_p \, \mathrm{Softmax}(T([I_d \quad 0_{d\times n}] X_p)^\top [I_d \quad 0_{d\times n}] X_p)$$
$$= X_p \, \mathrm{Softmax}(T X^\top X)$$

For each column in the above equation, we have

$$\mathrm{SA}_1(X_p)_{:,j} \qquad\qquad (\text{the } j\text{-th column})$$
$$= X_p \, \mathrm{Softmax}(T X^\top X)_{:,j}$$
$$= \sum_{i=1}^n \frac{e^{T x_i^\top x_j}}{\sum_{i'=1} e^{T x_{i'}^\top x_j}} \begin{bmatrix} x_i \\ r_i \end{bmatrix}, \qquad\qquad (\text{A.14})$$

where $r_i$ is the $i$-th column of the positional encoding $I_n$ as mentioned in the main text.

Because

$$x_i^\top x_j = -\frac{1}{2}(x_i - x_j)^\top (x_i - x_j) + \frac{1}{2}x_i^\top x_i + \frac{1}{2}x_j^\top x_j$$

$$= -\frac{1}{2}\|x_i - x_j\|_2^2 + \frac{1}{2}r_x^2 + \frac{1}{2}r_x^2 \qquad (r_x \text{ is in Assumption 3.2})$$

$$\le r_x^2$$

where the equality is only achieved when $\|x_i - x_j\|_2 = 0$, meaning $x_i = x_j$.

By Assumption 3.1, $x_i$ are from a finite dictionary. This mean there exists a $\Delta > 0$ such that for any two unequal $x_i, x_j \in \mathcal{D}$, they satisfy

$$\|x_i - x_j\|_2 \ge \Delta.$$

This means

$$x_i^\top x_j - r_x^2 = -\|x_i - x_j\|_2^2 \le -\Delta^2.$$

Because $\mathcal{D}$ is finite, we assume a bound on the infinite norm of its elements, denoted as $B_D$. This writes as

$$B_D = \sup_{x \in \mathcal{D}} (\|x\|_\infty).$$

Set $T$ to be

$$T \ge \frac{1}{\Delta^2} \ln\left(\frac{2nB_D}{\delta}\right).$$

Then we have for any $x_i \ne x_j$

$$\frac{e^{Tx_i^\top x_j}}{\sum_{i'=1} e^{Tx_{i'}^\top x_j}} \le \frac{e^{Tx_i^\top x_j}}{e^{Tx_j^\top x_j}} = e^{T(x_i^\top x_j - r_x^2)} \le e^{-T\Delta^2} \le \frac{\delta}{2nB_D}. \qquad (A.15)$$

We now verify the second condition on $\mathrm{SA}_1$ that is for $\delta$, we have

$$\|\mathrm{SA}_1(X_p)_{1:d,:} - X\|_\infty \le \delta$$

We verify this for each column. Specifically, we prove

$$\|\mathrm{SA}_1(X_p)_{1:d,j} - x_j\|_\infty \le \delta$$

for every $j \in [n]$.

By (A.14), the left-hand side of the above inequality is

$$\|\mathrm{SA}_1(X_p)_{1:d,j} - x_j\|_\infty = \|\sum_{i=1}^n \frac{e^{Tx_i^\top x_j}}{\sum_{i'=1} e^{Tx_{i'}^\top x_j}} \begin{bmatrix} x_i \\ r_i \end{bmatrix}_{1:d} - x_j\|_\infty$$

$$= \|\sum_{i=1}^n \frac{e^{Tx_i^\top x_j}}{\sum_{i'=1} e^{Tx_{i'}^\top x_j}} x_i - x_j\|_\infty$$

$$= \|\sum_{i=1}^n \frac{e^{Tx_i^\top x_j}}{\sum_{i'=1} e^{Tx_{i'}^\top x_j}} (x_i - x_j)\|_\infty$$

$$= \|\sum_{x_i \ne x_j} \frac{e^{Tx_i^\top x_j}}{\sum_{i'=1} e^{Tx_{i'}^\top x_j}} (x_i - x_j) + \sum_{x_i = x_j} \underbrace{\frac{e^{Tx_i^\top x_j}}{\sum_{i'=1} e^{Tx_{i'}^\top x_j}} (x_i - x_j)}_{0} \|_\infty$$

$$\le \sum_{x_i \ne x_j} \frac{e^{Tx_i^\top x_j}}{\sum_{i'=1} e^{Tx_{i'}^\top x_j}} \| \underbrace{(x_i - x_j)}_{\text{maximal entry smaller than } 2B_D} \|_\infty$$

$$\le ne^{-T\Delta^2} \cdot 2B_D \qquad (\text{By (A.15)})$$

$$\le \delta$$

**The Effect of** $SA_1$ **on Positional Encodings.** Since $SA_1$ preserves $X$, its main purpose is to alter the given positional encoding. Specifically, it **evens out the positional encoding of the identical tokens**.

For example, for an identical pair $x_{i_1} = x_{i_2}$, the positional encoding of both tokens will be arbitrarily close to $(r_{i_1} + r_{i_2})/2$ after the transformation of $SA_1$.

We verify this rigorously in the following discussions.

Let $r'_j$ denote the $j$-th positional encoding in the output of $SA_1$. From (A.14), we have

$$r'_j = \sum_{i=1}^{n} \frac{e^{T x_i^\top x_j}}{\sum_{i'=1} e^{T x_{i'}^\top x_j}} r_i.$$

From (A.15), set $T \geq 1/\Delta^2 \ln(2n/\delta)$, we have

$$\frac{e^{T x_i^\top x_j}}{\sum_{i'=1} e^{T x_{i'}^\top x_j}} \leq \frac{e^{T x_i^\top x_j}}{e^{T x_j^\top x_j}} = e^{T(x_i^\top x_j - r_x^2)} \leq e^{-T \Delta^2} \leq \frac{\delta}{2n}, \ x_i \neq x_j.$$

This means the proportion of $r_i$ whose corresponding $x_i \neq x_j$ is no larger than $n \cdot \delta/(2n) = \delta/2$.

Meanwhile for all $i$ that satisfies $x_i = x_j$, their proportion:

$$\frac{e^{T x_i^\top x_j}}{\sum_{i'=1} e^{T x_{i'}^\top x_j}}$$

are uniformly

$$\frac{e^{T x_j^\top x_j}}{\sum_{i'=1} e^{T x_{i'}^\top x_j}}.$$

This concludes $r'_j$ to be close to the average of all $r_i$ satisfying $x_i = x_j$ by an error (in infinite norm) no larger than

$$\frac{\delta}{2} \cdot \|r_i\|_1 \leq \frac{\delta}{2}. \tag{A.16}$$

We note again that $r_i$ is the $i$-th column of $I_n$.

In the following discussions, we first assume the input to the $F$ is

$$\begin{bmatrix} x_1 & x_2 & \cdots & x_n \\ r_1^* & r_1^* & \cdots & r_n^* \end{bmatrix}, \tag{A.17}$$

where $r_i^*$ is an average of all positional encodings whose connected token is identical to $x_i$. Previous discussions ((A.16)) denote the output of $SA_1$ to be arbitrarily close to the above input to $F$.

**Strategy of Information Extraction from** $F \circ SA_1(X_p) + X_p$. In order to explain our method of extracting information about identical tokens, we first need to prove some attributes of the output.

Let $F_i, i \in [N]$ denote the $i$-th layer in $F$. Let $W_K^i, W_Q^i$ and $W_V^i$ denote its $W_K, W_Q$ and $W_V$ matrices.

Let $Y_N$ denote the output of $F \circ SA_1(X_p)$, we divide $Y_N$ into two parts

- 1. $(Y_N)_{1:d,:}$, which is the first $d$ rows.

- 2. $(Y_N)_{d+1:d+n,:}$, which is the last $n$ rows.

For simplicity, In the following discussions, we use $Y_X$ to denote $(Y_N)_{1:d,:}$ and $Y_R$ to denote $(Y_N)_{d+1:d+n,:}$.

If two tokens $y_i, y_j$ in $Y_0$ are identical, then by Lemma A.4, $F_1(Y_0)_i = F_1(Y_0)_j$, which also writes as

$$(Y_1)_j = (Y_1)_j.$$

Repeating the above procedure yields

$$(Y_N)_i = (Y_N)_j.$$

**Our strategy** is as follows.

- Step 1. Extract $Y_R$.

- Step 2. Identify the identical tokens in $X + Y_X$.

- Step 3. The identical tokens in the output may not correspond to identical tokens in the input. This is only possible when the $Y_X$ compensated some differences between the different tokens in $X$.

- Step 4. Use $Y_R$ to filter out the exceptions in Step 3.

We then implement this strategy with the construction of $T_1$ and $T_2$.

**The Construction of $T_1$ and $T_2$.** We first construct a feed-forward layer $\text{FFN}_1$ to extract $Y_R$.

$$\text{FFN}_1(Z) := \underbrace{\begin{bmatrix} [I_d \quad 0_{d\times n}] Z \\ [0_{n\times d} \quad I_n] Z - (\frac{1}{\delta_1}\sigma([0_{n\times d} \quad I_n] Z + \delta_1 - 1) - \frac{1}{\delta_1}\sigma([0_{n\times d} \quad I_n] Z - 1)) \\ \frac{1}{\delta_1}\sigma([0_{n\times d} \quad I_n] Z + \delta_1 - 1) - \frac{1}{\delta_1}\sigma([0_{n\times d} \quad I_n] Z - 1) \end{bmatrix}}_{d+2n\times n}.$$

This also writes as

$$\text{FFN}_1(Z) = \underbrace{\begin{bmatrix} Z_{1:d,:} \\ Z_{d+1:d+n,:} - (\frac{1}{\delta_1}\sigma(Z_{d+1:d+n,:} + \delta_1 - 1) - \frac{1}{\delta_1}\sigma(Z_{d+1:d+n,:} - 1)) \\ \frac{1}{\delta_1}\sigma(Z_{d+1:d+n,:} + \delta_1 - 1) - \frac{1}{\delta_1}\sigma(Z_{d+1:d+n,:} - 1) \end{bmatrix}}_{d+2n\times n}.$$

Here $\delta_1$ is a small number we specify in later discussions.

When applied on $X_p + Y_N$, it outputs

$$\text{FFN}_1(X_p + Y_N) = \underbrace{\begin{bmatrix} X + Y_X \\ Y_R - (\frac{1}{\delta_1}\sigma(Y_R + I + \delta_1 - 1) - \frac{1}{\delta_1}\sigma(Y_R + I - 1)) \\ \frac{1}{\delta_1}\sigma(Y_R + I + I + \delta_1 - 1) - \frac{1}{\delta_1}\sigma(Y_R + I - 1) \end{bmatrix}}_{d+2n\times n}.$$

Here we use adding a constant to a matrix to denote adding this constant to *every entry* of the matrix.

Let $S_i$ denote the attention score matrix of the $i$-th layer in $F$, this means

$$S_i := \text{Softmax}((W_K^i Y_{i-1})^\top (W_Q^i Y_{i-1})),$$

We note that

$$\frac{1}{\delta_1}\sigma(Y_R + I + \delta_1 - 1) - \frac{1}{\delta_1}\sigma(Y_R + I - 1) = I. \tag{A.18}$$

This is due to

$$\|Y_R\|_\infty < \|Y_R\|_1$$

$$= \|\prod_{i=1}^{N}(W_4^i \cdot I \cdot S_i)\|_1$$

$$\leq \|\prod_{i=1}^{N} W_4^i\|_1 \qquad (S_i \text{ performs a weighted average})$$

$$\leq \prod_{i=1}^{N} \|W_4^i\|_{1,1}$$

$$\leq 1.$$

Here $W_4^i$ denotes $(W_V^i)_{d+1:d+n,d+1:d+n}$

This means for the $i$-th column, all entries beside the $i$-th one is smaller than 1. Then there is a $\delta_1 > 0$ such that they are uniformly smaller than $1 - \delta_1$. That yields us the result in (A.18).

Thus the output of $\text{FFN}_1$ is

$$\begin{bmatrix} X + Y_X \\ Y_R + I - I \\ I \end{bmatrix} = \begin{bmatrix} X + Y_X \\ Y_R \\ I \end{bmatrix}$$

If $x_i = x_j$, by Lemma A.4, $(Y_N)_i = (Y_N)_j$, meaning

$$\begin{bmatrix} X + Y_X \\ Y_R \end{bmatrix}_i = \begin{bmatrix} X + Y_X \\ Y_R \end{bmatrix}_j$$

When facing the opposite situation, $x_i \neq x_j$, if

$$\begin{bmatrix} X + Y_X \\ Y_R \end{bmatrix}_i = \begin{bmatrix} X + Y_X \\ Y_R \end{bmatrix}_j$$

Then $(Y_X)_i \neq (Y_X)_j$, yet $(Y_R)_i = (Y_R)_j$

As previously stated

$$Y_R = \prod_{i=1}^{N} (W_2^i \cdot I \cdot S_i),$$

which means

$$\prod_{i=1}^{N} (W_2^i)^{-1} Y_R = \prod_{i=1}^{N} S_i$$

Because the $i,j$-th entry of $\prod_{i=1}^{N} S_i$ denotes how much of $(Y_N)_j$ is comprised of $\prod_{i=1}^{N} W_1^i x_i$.

Therefore $(Y_R)_i = (Y_R)_j$ infers $(Y_X)_i = (Y_X)_j$, this contradicts with the assumption of

$$\begin{bmatrix} X + Y_X \\ Y_R \end{bmatrix}_i = \begin{bmatrix} X + Y_X \\ Y_R \end{bmatrix}_j.$$

And hence the identity between the tokens in $X$ is strictly reflected by

$$\begin{bmatrix} X + Y_X \\ Y_R \end{bmatrix}$$

Then by Theorem 3.2, there exists two single-head attentions $\text{SA}_2$ and $\text{SA}_3$ and a feed-forward network $\text{FFN}_2$ such that

$$\text{FFN}_2 \circ \text{SA}_3 \circ \text{SA}_2(\text{FFN}_1(X_p + Y_N)) A_c$$

is arbitrarily close to an induction head on $X + Y_X$, which is $\text{LaIND}_{F \circ \text{SA} + I}$.

Finally, let $L$ denote the Lipschitzness of the whole network $(\text{FFN}_2 \circ \text{SA}_3 \circ \text{SA}_2(\text{FFN}_1(X_p + Y_N)) A_c)$ we constructed. Because this network is continuous, $L$ exists.

Then, since our discussion revolves around the assumed input in (A.17), which has a distance arbitrarily close to the actual input (when configuring $\text{SA}_1$ to be of a specific form).

The actual output has a distance to the $\text{FFN}_2 \circ \text{SA}_3 \circ \text{SA}_2(\text{FFN}_1(X_p + Y_N)) A_c$ that is at most $L$ times the distance between the actual input and the assumed input, which is also arbitrarily small.

This completes our proof.

$\square$

## B    INDUCTION HEAD OF ANOTHER DEFINITION

Though the induction head in the main text is defined as the average of all tokens next to the tokens identical to $x_n$, this form is not mandatory in our theory, as another form of induction heads yields a similar result

- Induction head that outputs the token next to the last (or the first) token that's identical to $x_n$.

This writes out as

**Definition B.1** (Alternative Definition of Induction Head). For an input sequence $X = [x_1 \, x_2 \, \cdots \, x_n]$, define the induction head IND to be

$$\mathrm{IND}(X) = x_{k+1},$$

where $k = \max_{x_i = x_n, i < n}(i)$.

We give this result as the following corollary of Theorem 3.2.

**Corollary B.0.1** (Induction Head Taking the Last Identical Token). There exist two single-head attention blocks $\mathrm{SA}_1$ and $\mathrm{SA}_2$ connected by a feed-forward network FFN and an $A_c \in \mathbb{R}^n$, where $\mathrm{SA}_1, A_c$ only need to satisfy the conditions for $\mathrm{SA}_1$ in Theorem 3.2, such that

$$\|\mathrm{SA}_2 \circ \mathrm{FFN} \circ \mathrm{SA}_1(X)A_c - \mathrm{IND}(X)\|_\infty \le \epsilon$$

for any $\epsilon > 0$.

*Proof.* Let $\mathrm{SA}_2^*, \mathrm{SA}_1^*, \mathrm{FFN}^*$ denote a feasible construction for Theorem 3.2 (that is, a feasible construction for induction head defined in Definition A.3).

Let $\mathrm{SA}_1$ be $\mathrm{SA}_1^*$, and let FFN be

$$\mathrm{FFN} = \begin{bmatrix} 0 \\ 1 \\ \vdots \\ n-1 \end{bmatrix} \odot \mathrm{FFN}^*.$$

Let $\mathrm{SA}_2$ be $\mathrm{SA}_2^*$, the attention score matrix of $\mathrm{SA}_2$ is that of $\mathrm{SA}_2^*$ with all $\mathrm{ID}(X)$ replaced as $c \odot \mathrm{ID}(X)$ in which $c$ is

$$\begin{bmatrix} 0 \\ 1 \\ \vdots \\ n-1 \end{bmatrix}.$$

This yields the attention score matrix of $\mathrm{SA}_2$ to be

$$\frac{1}{\sqrt{m}}R^\top R + \sqrt{m}c \odot \mathrm{ID}(X) \cdot 1_{1\times d} \to \sqrt{m}c \odot \mathrm{ID}(X) \cdot 1_{1\times d}$$

when $m \to +\infty$.

For any $\epsilon_0 > 0$, when $c$ is sufficiently large, Softmax approximates a hard max function which gives

$$\|\mathrm{Softmax}(\frac{1}{\sqrt{m}}R^\top R + \sqrt{m}c \odot \mathrm{ID}(X))_{:,c} - e_{\max_i(\frac{1}{\sqrt{m}}R^\top R_{i,c} + c_i \mathrm{ID}(X)_i)}\|_\infty \le \epsilon_1,$$

which is equivalent to

$$\|\mathrm{Softmax}(\frac{1}{\sqrt{m}}R^\top R + \sqrt{m}c \odot \mathrm{ID}(X))_{:,c} - e_{\max_i(c_i \mathrm{ID}(X)_i)}\|_\infty \le \epsilon_1,$$

when $m$ is sufficiently large. Here $e_k$ denotes a $n$-dimensional one-hot vector (a vector with only one entry being 1 and others being 0) whose non-zero entry is the $k$-th.

**Remark B.1.** Softmax approximating hard max has a prerequisite that the operating tensor's maximal entry always has a difference with the second largest entry for a constant that's not 0. Because here the maximal entry of $c \odot \mathrm{ID}(X)$ (when it's not all-zero) is at least larger than the second largest by 1, and $1/m R^\top R$ goes to 0, we are able to ensure a minimal between the largest and second largest entry of every column in

$$\frac{1}{\sqrt{m}} R^\top R + \sqrt{m} c \odot \mathrm{ID}(X).$$

Because $\max_i(c_i \mathrm{ID}(X)_i)$ denotes the label of the last token in $X$ that's identical $x_n$, and that $\mathrm{SA}_2$ multiplies this with the original $X$, this extracts $x_{\max_i(c_i \mathrm{ID}(X)_i)}$ and yield the final result.

$\square$

## C  NUMERICAL EXPERIMENT

To examine the validity of our construction, we implement our construction in Python and randomly set the part of the weight in the first layer of attention according to Theorem 3.2. For simplicity of code implementation, the positional encoding is fixed to be the identity matrix, as well as $W_V$ in the first layer of attention.

**Objective.** This experiment aims to verify if a two-layer attention network connected by a feed-forward network, whose first layer of attention has random weights according to Theorem 3.2, is able to implement an induction head.

**Experiment Setting.** We implement an induction head with part of its weights being random according to our construction in Theorem 3.2. We consider the input sequences of length 8 and token length 10. For simplicity, we fix the repeated token to be the first one. The first layer of attention has a hidden dimension of 100. We run the process of

- First, randomly set part of the weights in the first attention layer.
- Second, randomly set the part in the input that's not repeated.
- Finally, calculate the absolute difference between the output of the constructed network and the ground truth.

for 1000 times and average their loss as the final output.

**Results.** After 50 rounds of experiments, we observe a total average difference of 0.14. We draw a scatter plot of all results in Figure 2.

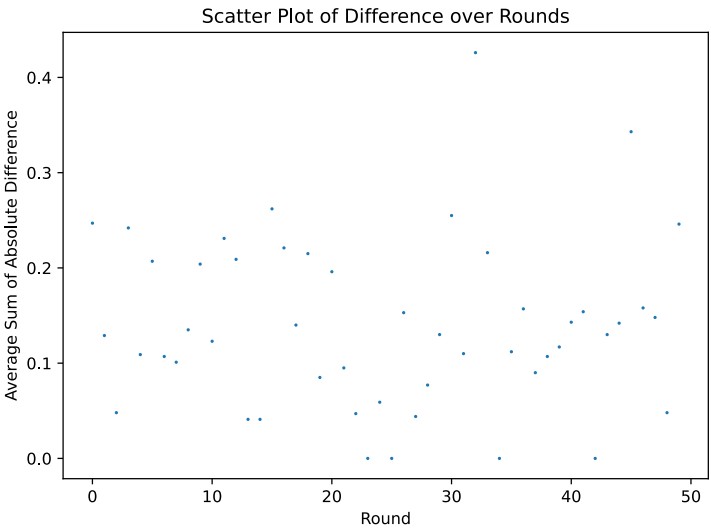

Figure 2: The result of the experiment in Appendix C, every dot denotes the average of the absolute difference between the model output and the ground truth for an induction head.

