# OpenReview forum: "Induction Head Implementation Across Diverse Transformer Weight Constructions"
_ICLR.cc/2026/Conference — Submitted to ICLR 2026_

### Official Review · Reviewer_WYnG · 2025-10-31

**Soundness:** 2
**Presentation:** 3
**Contribution:** 2
**Rating:** 2
**Confidence:** 3

**Summary:**

This paper systematically investigates the implementation of induction heads across diverse Transformer architectures, proposing a generic construction that does not rely on relative positional encodings, and extends the theoretical construction for induction heads in multi-layer networks.

**Strengths:**

1. The paper removes reliance on relative position encoding and proposes an alternative method for constructing induction heads.

2. The synthetic experiment empirically supports the theoretical construction.

**Weaknesses:**

1. (main) Although the authors propose new constructions of induction heads, it remains unclear whether these constructions are realized in practice. The experiments are highly synthetic and do not demonstrate that induction heads learned by transformers on real datasets conform to the proposed constructions, which limits the paper’s significance. I **strongly recommend** verifying this point by training transformers on NLP datasets and probing the resulting models.

2. From a theoretical perspective, the work presents only a construction; it remains unclear how gradient descent would discover these configurations in practice.

3. It is unsurprising that induction heads can be achieved using alternative positional encodings, as induction heads were first identified in GPT models employing absolute positional encodings.

4. Experimental details are incomplete, including information about the data distribution.

**Questions:**

See Weakness.

---

### Official Review · Reviewer_GBxZ · 2025-11-01

**Soundness:** 2
**Presentation:** 2
**Contribution:** 2
**Rating:** 4
**Confidence:** 4

**Summary:**

The paper gives a theoretical account of how transformers implement induction heads — the mechanism that lets a model see a pattern ... A B ... A and then predict B next. The following is the core results of the paper:

1. Transformers can implement induction. A small, standard transformer (just two layers) can learn the classic induction behavior. This does not require special positional encodings or carefully hand-crafted weights.The same basic induction behavior can emerge in normal multi-head / multi-layer transformers. It also works in deep models. Even in later layers, where tokens have already been transformed, the model can still perform an induction-like lookup — copying not the raw next token, but the next token’s representation. So the mechanism scales with depth.

2. The authors show that in synthetic experiments, partially frozen transformers still learn this induction behavior and become highly accurate, which supports the claim that it’s robust and naturally learnable.

**Strengths:**

The paper provides an alternative to the standard induction head mechanism. While most prior work models induction as copying parent tokens and comparing them to derive the prediction target, this paper introduces a different perspective: each token position produces an indicator vector encoding alignment with the final token, which is then aggregated via a second attention layer. The paper also includes a proof-of-concept experiment.

**Weaknesses:**

While the paper offers an interesting theoretical construction, several aspects remain unclear or potentially unrealistic:

1. Unrealistic global access assumption. The construction in Theorem 3.1 appears to require each position to aggregate global information about the sequence, including knowledge of the last token. This assumption is incompatible with causal attention, which restricts each position to past context. The paper does not discuss how such global aggregation could be approximated or implemented in a causal transformer.

2. Scalability and practicality. Even setting aside causality, the mechanism in Theorem 3.1 assumes precise sequence-wide coordination that seems impractical for long contexts. It is unclear whether such an exact global signal can emerge or be learned in realistic training regimes.

3. Connection to real models. The paper would benefit from stronger theoretical or empirical evidence that this form of “indicator-based induction” actually arises in trained transformers. Compared with known induction-head implementations, this construction appears more complex and less plausible as a naturally learned mechanism.

Overall, while the theoretical result is neat, its relevance to practical transformer behavior remains uncertain without further justification or evidence.

**Questions:**

No further questions

---

### Official Review · Reviewer_i8Ud · 2025-11-01

**Soundness:** 3
**Presentation:** 3
**Contribution:** 2
**Rating:** 4
**Confidence:** 4

**Summary:**

This paper presents a theoretical analysis of how induction heads can be implemented in a wide range of transformer architectures. The authors provide constructive proofs to show that these mechanisms are not rigid circuits but flexible components that can emerge from broad weight configurations, notably removing the reliance on specific relative positional embeddings. They propose a novel, inverted match-then-copy circuit and extend their analysis to multi-layer settings by introducing the concept of a "latent space induction head." The work's central idea is that the capability to form induction heads is a more general and adaptable feature of transformers, rather than being tied to a single, specific implementation.

**Strengths:**

*   **Novel Construction for Induction heads:** The paper presents a novel and interesting constructive proof for how induction heads can be realized in transformers.
*   **Generalization of Positional Encodings:** It generalizes prior work by demonstrating that induction heads can be formed with any full-rank absolute positional encoding, removing the reliance on relative positional embeddings.
*   **Flexibility of Induction Heads:** The work mainly argues that induction heads are not a single, rigid circuit but a flexible phenomenon that can emerge from a wide class of model weights.

**Weaknesses:**

**Gap Between representation and learning:** The paper provides an interesting existence proof, demonstrating that a wide class of Transformer weights can indeed represent induction heads. However, the core of the analysis focuses on what a network can represent, without any evidence that such structures actually emerge through optimization and therefore are realistic to study. It's unclear to me if these elegant theoretical solutions are on a plausible optimization path for a network trained with gradient descent, or if they represent one of many possible, but not necessarily learned, solutions.

**Potential issue in the proof of Theorem A.1.** The target vector ID(X) is defined with a zero in its first entry; however, the construction seems to me to produce a 1 in that position. Specifically, the transformation matrix E ensures the first row of the intermediate matrix Z* is zero. According to the FFN's definition and the paper's own calculation, a zero input yields an output of 1. This would make the first row of the final output matrix a row of ones, which seems to contradict the definition of ID(X).

**Relative positional encoding:** The paper proposes a new, inverted mechanism for induction heads where Layer 1 performs the token matching and Layer 2 performs the copy. While this is a valid theoretical construction, I wonder if it is less flexible than standard induction heads. The "+1" offset logic (copying the token immediately following a match) seems to me hardcoded into the architecture of the second layer's attention and FFN. This contrasts with mechanisms that utilize relative positional information, which can more naturally learn to attend to various offsets.

**Multi-Head Case :** The construction seems to reduce the multi-head problem to the single-head case and does not seem to address the more complex question of how the output from multiple heads is used to form an induction circuit.

**Experiments:** The experiments lack section lacks clarity about the training task. It is unclear if the repeated pattern is newly generated for each sequence, which would force the model to perform in-context learning via an induction mechanism or if the patterns are drawn from a fixed set, the model could solve the task by simply memorizing associations without needing to look back at the context, which would not necessarily validate the formation of induction heads.

Furthermore, the experiment, which trains a network with some layers randomly frozen, demonstrates a degree of flexibility but does not directly test whether the specific, multi-stage circuit proposed in the theoretical proofs is what a model actually learns. The experiment supports the general idea that induction can emerge under constraints, but it doesn't bridge the gap between the paper's specific construction and the mechanisms learned in practice.

**Questions:**

2. Could the authors please clarify the apparent discrepancy between the constructionand ID(X)? Is there a step in the FFN construction or the preceding transformations that ensures the first element of the final output is zero, consistent with the definition of ID(X)?

3. Could the authors comment on the flexibility of their proposed two-layer circuit? For instance, could the model learn to implement an induction head that copies the token at position i+2 instead of i+1, or would this require a completely different weight construction for the second layer?

4. The paper highlights removing the reliance on RPE as a key contribution. However, some prior theoretical work on induction heads also uses absolute positional encodings. Could the authors clarify what specific "reliance on RPE" in previous research their work overcomes? Does this refer to a specific architectural RPE component, or the conceptual reliance on relative distances between tokens, which their model also seems to bypass in favor of an absolute-indexing copy mechanism?

5. Could the authors please clarify the data generation process for the experiment in Section 4? Specifically, is the A B C -> D pattern newly and randomly generated for each sequence in the training set, or are the patterns drawn from a fixed, predefined set?

6. Given that the proposed theoretical construction is quite specific, what evidence from this experiment suggests that the trained model is implementing this particular circuit, as opposed to a more standard induction head mechanism?

---

### Official Review · Reviewer_nKn1 · 2025-11-05

**Soundness:** 4
**Presentation:** 3
**Contribution:** 3
**Rating:** 8
**Confidence:** 4

**Summary:**

This paper focuses on the implementation flexibility and universality of induction heads—key self-attention mechanisms enabling in-context learning (ICL) in transformers—across diverse transformer architectures. Unlike prior work that relied on specific designs (e.g., relative positional embeddings, rigid circuit structures) to study induction heads, the authors propose a new theoretical framework showing that:
1. Two-layer induction heads exhibit significant flexibility in their first-layer weight construction, allowing compatibility with other network modules;
2. A "latent space induction head" mechanism can preserve token identity information and function in deep multi-layer networks where retrieving original inputs is challenging;
3. Induction heads can be trained using only a subset of model layers. The authors validate their theory with proof-of-concept experiments on 4-layer and 6-layer transformers, showing that models with partially fixed weights still achieve over 83% accuracy on a synthetic ICL task.

**Strengths:**

- Theoretical Novelty: The framework’s ability to generalize induction heads to diverse architectures (single-head/multi-head, shallow/deep) fills a critical gap in prior work, which often focused on rigid circuit designs. The proof that "token identity information is robust under transformer transformations" (Section 3.4) is a key insight for understanding ICL in deep models.

- Rigorous Mathematical Foundations: The appendices provide complete, detailed proofs (e.g., Lemma A.2 on attention preserving identity, Theorem A.4 on latent space induction heads) that enable other researchers to build on the work—essential for theoretical ML papers.

- Targeted Experiments: The experiments directly test the core claim of "flexible weight construction": training only the 1st/4th layers (and input/output layers) while fixing others still yields high accuracy (>83%), validating that induction heads do not require full model training.

**Weaknesses:**

Idealized Assumptions: Key assumptions (fixed input token norms, block-diagonal $W_V$) exclude common LLM settings: open-vocabulary settings, dynamic token norms, and low-rank $W_V$ (used for efficiency in large models). The paper acknowledges these limitations but does not discuss how to relax them.

Lack of Causal Interventions: The experiments measure accuracy but do not perform causal tests (e.g., ablating the proposed latent space induction head, perturbing token identity information) to confirm that the observed performance gains are indeed due to induction heads—unlike prior work (e.g., Crosbie & Shutova, 2024) that ablated induction heads to show ICL degradation.

**Questions:**

1. In Section 3.4, you assume block-diagonal $W_V$ to avoid mixing content and positional encoding. Do you have preliminary results or intuitions about how relaxing this assumption (e.g., allowing partial mixing of content and position) would affect the latent space induction head’s performance? Would mixing break token identity preservation?

2. The experiments train only the 1st and 4th layers. Did you sweep which layers to train (e.g., 2nd/5th, 3rd/6th) to test if induction head training is sensitive to layer choice? If induction heads are truly flexible, performance should be consistent across layer pairs—but this is not verified.

3. You mention that larger models shift to function-vector heads (Yin & Steinhardt, 2025). Do you have any insights into how your latent space induction head interacts with function-vector heads? For example, could induction heads bootstrap function-vector heads in deep models, as suggested in the related work?

---

### Meta-Review · Area_Chair_ffiY · 2026-01-15

**Summary:**

The paper proposes a theoretical framework demonstrating the flexibility of induction heads across diverse transformer architectures, effectively removing the reliance on specific relative positional embeddings (RPE). Reviewers generally appreciated the theoretical novelty and the rigorous mathematical approach to constructing these heads in various settings (single/multi-head, shallow/deep).

However, significant concerns were raised regarding the soundness and practicality of the theory. Specifically, Reviewer GBxZ identified a potential violation of causal attention in Theorem 3.1, noting that the construction appears to require global access to the sequence. Reviewer i8Ud pointed out a potential mathematical contradiction in Theorem A.1 regarding the definition of the identity vector. Furthermore, multiple reviewers (i8Ud, WYnG) highlighted a gap between the theoretical existence of these weight configurations and whether such circuits are actually learned via gradient descent, noting that the synthetic experiments did not sufficiently bridge this gap or perform causal interventions to verify the mechanism.

**Reviewer Concerns:**

As the authors did not submit a rebuttal, no reviewer concerns were addressed.

Outstanding Concerns:
- Causality Violation (Theorem 3.1): Reviewer GBxZ's concern that the construction requires global sequence aggregation (violating the autoregressive nature of causal transformers) remains unaddressed. This is a potential fatal flaw for the theoretical claims.
- Mathematical Correctness (Theorem A.1): Reviewer i8Ud's specific query regarding the discrepancy between the target vector definition ($ID(X)$ having a zero first entry) and the construction yielding a one remains unresolved.
- Existence vs. Learning Gap: The core concern shared by i8Ud and WYnG—that proving a weight configuration exists does not prove a model will learn it—remains a significant hurdle. The lack of causal interventions (ablations) in the experiments leaves this open.
- Idealized Assumptions: The reliance on block-diagonal matrices and fixed token norms (Reviewer nKn1) was not justified or relaxed.

**Reviewer Scores:**

The authors did not engage in discussion.

---

### Decision · Program_Chairs · 2026-01-26

Reject